# Recent Advances in the Development of Nanodelivery Systems Targeting the TRAIL Death Receptor Pathway

**DOI:** 10.3390/pharmaceutics15020515

**Published:** 2023-02-03

**Authors:** Anne V. Yagolovich, Marine E. Gasparian, Dmitry A. Dolgikh

**Affiliations:** 1Shemyakin-Ovchinnikov Institute of Bioorganic Chemistry of the Russian Academy of Sciences, 117997 Moscow, Russia; 2Faculty of Biology, Lomonosov Moscow State University, 119234 Moscow, Russia

**Keywords:** TRAIL, death receptors, DR5, receptor clustering, ligand-targeted drugs, drug delivery, nanoparticles, nanotherapeutics

## Abstract

The TRAIL (TNF-related apoptosis-inducing ligand) apoptotic pathway is extensively exploited in the development of targeted antitumor therapy due to TRAIL specificity towards its cognate receptors, namely death receptors DR4 and DR5. Although therapies targeting the TRAIL pathway have encountered many obstacles in attempts at clinical implementation for cancer treatment, the unique features of the TRAIL signaling pathway continue to attract the attention of researchers. Special attention is paid to the design of novel nanoscaled delivery systems, primarily aimed at increasing the valency of the ligand for improved death receptor clustering that enhances apoptotic signaling. Optionally, complex nanoformulations can allow the encapsulation of several therapeutic molecules for a combined synergistic effect, for example, chemotherapeutic agents or photosensitizers. Scaffolds for the developed nanodelivery systems are fabricated by a wide range of conventional clinically approved materials and innovative ones, including metals, carbon, lipids, polymers, nanogels, protein nanocages, virus-based nanoparticles, dendrimers, DNA origami nanostructures, and their complex combinations. Most nanotherapeutics targeting the TRAIL pathway are aimed at tumor therapy and theranostics. However, given the wide spectrum of action of TRAIL due to its natural role in immune system homeostasis, other therapeutic areas are also involved, such as liver fibrosis, rheumatoid arthritis, Alzheimer’s disease, and inflammatory diseases caused by bacterial infections. This review summarizes the recent innovative developments in the design of nanodelivery systems modified with TRAIL pathway-targeting ligands.

## 1. Introduction

The main goal of nanotherapeutic drug development has long been to overcome the limitations of conventional drugs, mainly for cancer treatment, by improving bioavailability, pharmacokinetics, and targeted drug delivery. Several dozen nanoformulations have been approved to date for clinical use by the Food and Drug Administration (FDA) and the European Medicines Agency (EMA), and many are under clinical investigation. However, the majority of approved nanomedicines obtained a lowered toxicity rather than enhanced efficacy. Recent innovations in nanomaterial science aim to create new combinatorial strategies and complex designs to develop novel and effective targeted nanotherapeutics. After addressing important issues, such as efficacy, safety, reproducibility at commercial-scale manufacturing, and regulatory frameworks, the novel complex nanomedicine formulations may evolve into innovative targeted therapeutic strategies [1,2].

TRAIL (TNF-related apoptosis-inducing ligand) is a naturally occurring cytokine from the TNF (tumor necrosis factor) superfamily, which is produced by most normal cells but primarily by dendritic cells and natural killer (NK) cells. TRAIL has an outstanding ability to selectively induce apoptosis in tumor cells upon binding to its cognate death receptors, DR4 or DR5 (Figure 1). In this regard, TRAIL has been intensely investigated as a potential antitumor agent but has shown insufficient efficacy in subsequent clinical trials. Due to its unique tumor selectivity and cell-killing capacity, the TRAIL signaling pathway continues to attract the attention of researchers. Thus, further works have focused on enhancing its antitumor properties [3].

Targeting the TRAIL pathway by nanocarriers primarily pursues two goals: increasing the valency of the ligand for more efficient clustering of death receptors DR4 and/or DR5 and additional passive targeting to the lesion site, which is an inherent function of many types of nanoparticles. Activation of TRAIL death receptors may be accomplished either by the recombinant TRAIL protein and its modified variants, anti-death receptor agonistic antibodies, or specially designed TRAIL mimicking peptides. Additionally, if TRAIL itself or rather small peptides are used as a targeting ligand, conjugation with nanoparticles aims to improve the initially poor pharmacokinetic parameters. Additionally, since nanocarriers are usually complex nanoscale systems often composed of several materials with opposite physicochemical properties, they are able to simultaneously deliver other important therapeutic molecules obtaining a synergistic effect. Thus, nanodelivery provides a combination of crucial therapeutic advantages, including both active and passive targeting of a lesion, the ability to encapsulate molecules with opposite properties, improved solubility for non-water-soluble molecules, a better safety profile for toxic drugs, and prolonged and controllable drug release.

TRAIL death receptors DR4 and DR5 are both able to trigger programmed cell death upon ligand binding. Despite the rise of controversial evidence on the pivotal role of the DR4 receptor in apoptosis induction by TRAIL in some types of cancer [4,5], the DR5 receptor is traditionally considered more potent and crucial for triggering programmed cell death upon TRAIL binding. For example, DR5 expression is correlated with lymph node metastases and poor survival in breast cancer [6], and targeting DR5 suppressed breast cancer metastases more effectively than targeting DR4 [7]. Importantly, DR4 expression is generally less common when compared with DR5, which is prevalent and overexpressed in most types of cancer. For these reasons, the majority of TRAIL pathway-targeted nanocarriers are aimed at either DR5 alone or both DR4 and DR5. Ultimately, all of them pursue to induce efficient, coordinated death receptor clustering, resulting in increased apoptosis signaling and enhanced therapeutic efficiency.

The development of therapeutic approaches for targeting the TRAIL pathway was comprehensively reviewed a few years ago [8,9,10,11,12]. However, significant research progress over recent years in the area of TRAIL-related nanodelivery systems urges summarization of the new data for a better understanding of the relevant developments in the field of nanosized therapeutics targeting the TRAIL pathway. Further, important studies may have escaped the attention of the previous reviewers. This narrative review extensively surveys all types of synthetic nanodelivery systems bearing TRAIL pathway-targeting ligands developed over the past decade. The criteria for the inclusion of delivery systems in the review were the nanosize and functionalization with a ligand targeting one of the key activators of the TRAIL pathway in order to achieve enhanced therapeutic potential. Special attention was paid to the fabrication techniques and mechanism of action of the obtained nanocarriers, modified with targeted protein ligands. A detailed description of the methods and the obtained results may help assess the feasibility of a particular area of research. Importantly, the current review not only covers a broad spectrum of TRAIL-related delivery systems that have recently been developed for cancer treatment, which is considered conventional for TRAIL, but it also includes those aiming at other therapeutic areas, such as autoimmune and inflammatory diseases, due to TRAIL’s important role in the regulation of immune processes.

## 2. Inorganic Nanoparticles Modified with TRAIL Pathway-Targeting Ligands

Inorganic nanomaterials possess unique size-dependent optical, magnetic, electronic, and catalytic properties that determine their specific biological behavior. Such physicochemical versatility, tunable composition, high stability, and large surface area make them an emerging drug delivery system for various biomedical applications [13]. Over the past decade, a variety of inorganic nanoparticles were developed for the delivery of TRAIL pathway-targeting ligands.

### 2.1. Metal-Based Nanoparticles

#### 2.1.1. Gold and Silver Nanoparticles

Gold nanoparticles with controlled geometrical, optical, and surface chemical properties are the subject of intense research and applications in biology and medicine, particularly for the delivery of target molecules [14]. They are suitable for modification with different substances, which broadens their range of applications. For example, for targeting the TRAIL death receptor pathway, multi-functional gold nanoparticles with an outer lipid layer comprising of free carboxyl groups were encapsulated with oxaliplatin and surface modified by a DR5-targeted antibody containing an amine group via carbodiimide chemistry. The obtained immuno-gold nanoparticles (Co-Ox-AuNPs) synergistically reduced xenograft HCT-116 tumor growth by dual active and passive targeting [15]. In a similar approach, citrate-coated nanogolds were conjugated with the recombinant human TRAIL protein to develop nanogold-TRAIL complexes, which promote death receptor activation in M2 and tumor-associated macrophages (TAMs), displaying anti-inflammatory and pro-tumorigenic activity. The obtained nanogold-TRAIL complexes selectively increased the cytotoxicity of TRAIL by changing O-glycosylation levels in M2-polarized macrophages while remaining nontoxic to M1 macrophages and normal cells. This finding could help develop new nanomedicines for TAM-based cancer treatment [16].

Similarly, silver nanoparticles (AgNPs) have exclusive properties, which can be useful for antimicrobial applications, biosensor materials, composite fibers, cryogenic superconducting materials, cosmetic products, and the pharmaceutical industry [17]. To absorb TRAIL protein non-covalently, AgNPs have been citrate-reduced by approximately the same technique as with the abovementioned nanogolds. The obtained TRAIL-AgNPs appeared more toxic, compared to TRAIL and AgNPs alone, to TRAIL-resistant derivatives of human glioblastoma T98G cells by increasing the caspase activity [18]. In more recent work, a silver nanoparticle system containing cysteine as a functional group was conjugated with TRAIL by hydrogen bonds and coated with PEG to form AgCTP NPs, which inhibited the proliferation and colony formation of colon cancer HT-29 cells. The behavior of AgCTP NPs under physiological conditions, namely, non-toxicity and low hemolysis rate, was promising for in vivo applications [19].

#### 2.1.2. Iron Oxide Nanoparticles

Iron oxide nanoparticles are the most studied among the FDA-approved nanomedicines and have proved to be effective as contrast agents, drug delivery vehicles, and thermal-based therapeutics, appreciable for a variety of biomedical applications, such as diagnostics, imaging, and photothermal therapies. The iron oxide NP’s biocompatibility is assured by natural occurrence in human cells and a remarkable advantage of possible neosynthesis *in cellulo* [20]. In general, they possess acceptable biocompatibility and stability; however, they are restricted for some clinical applications due to low solubility and toxicity effects [21]. Importantly, the magnetocalorimetric properties of the iron oxide NPs, namely, the ability to induce heating by magnetic hyperthermia (MHT) and photothermia (PT) [22], can be exploited for sensitization to TRAIL-dependent cell death. 

It was previously shown that c-FLIP is a thermosensitive protein whose targeting by hyperthermia allows for the restoration of TRAIL-induced apoptosis [23]. To apply this strategy, TRAIL was grafted onto iron oxide nanoclusters (NCs) with the aim of increasing its pro-apoptotic potential through nanoparticle-mediated MHT or PT processes. NCs were then functionalized with (3-aminopropyl)triethoxysilane (APTES) to put amino groups onto their surface, followed by grafting of TRAIL onto the NCs by the formation of an amide bond between the amino groups and the carboxylic acid groups of TRAIL. The obtained NC@TRAIL induced breast cancer MDA-MB-231 cell death by the interaction of the NC with TRAIL receptors DR4 and DR5 during thermal treatment. NC@TRAIL initiated heating by either MHT or PT generated hotspots around the nanoclusters and, therefore, at the cell surface in the vicinity of the targeted receptors DR4 and DR5, leading to disruption of the membrane and subsequent cell death. The development of such a magnetothermal and photothermal nanoheater can pave the way to remote-controlled antitumor-targeted thermal therapies [24]. 

Additionally, the authors investigated whether the size of the nanovectors affected the efficiency and selectivity of TRAIL. For this, TRAIL was grafted onto magnetic spinel iron oxide NPs of defined core size. As a result, 100-nm NPs (NV100) showed higher pro-apoptotic activity than 10-nm NPs (NV10). This indicates that the size of the NPs is important upon TRAIL vectorization, and the apoptotic potential of the NPs can be governed by changing their size [25]. This work has recently been continued by exploring the methods of coupling iron oxide nanovectors with TRAIL to reveal the optimal nanohybrid structure. TRAIL was grafted to maghemite NPs in two different ways: by using carboxylic acid groups of the protein to graft it onto maghemite NPs previously functionalized with amino groups and by using the amino functions of the protein to graft it onto NPs functionalized with carboxylic acid groups. The pro-apoptotic effect of the former nanovector, named NH-TRAIL@NPs-CO, was greater than the latter, CO-TRAIL@NPs-NH, in human breast and lung carcinoma cells. It is worth mentioning that the computational study indicated that regardless of whether TRAIL was attached to NPs through acid or amino groups, DR4 recognition was not affected in either case [26]. This is consistent with much evidence on the major role of DR5, but not DR4, receptors in transmitting the apoptosis signal in most cancer cell lines. Therefore, further computational studies of NPs with DR5 receptors would be desirable.

Another research group has developed an iron oxide cluster-based nanoplatform (NanoTRAIL), where TRAIL was immobilized by spontaneously self-assembling via electrostatic interactions with a surface modified by positively charged stearic acid. The iron oxide NPs in the NanoTRAIL formulation initiated oxidative stress, followed by reactive oxygen species (ROS)-triggered c-Jun N-terminal kinase (JNK) activation and subsequent autophagy-assisted DR5 upregulation, resulting in a significantly improved survival outcome in a colorectal cancer patient-derived xenograft model compared with TRAIL monotherapy. Here, ROS generation by Fe^2+^ (Fenton reaction pathway) can serve as a prospective alternative to the chemically reactive ROS Inducers [27].

### 2.2. Carbon-Based Nanoparticles

#### 2.2.1. Graphene Scaffolds

Graphene is a two-dimensional atomic-scale graphite with sp2-hybridized carbon atoms arranged in a honeycombed alignment. Graphene-based materials are attractive for biomedical applications due to their physicochemical and biological properties, such as large surface area, chemical and mechanical stability, high thermal conductivity, and biocompatibility. Graphene and its derivatives can be conjugated with other aromatic materials on the surface through π–π stacking or electrostatic interactions, and the large surface area of graphene increases the opportunities for multi-drug delivery [28]. In one of the first works in the field of graphene-based targeting of the TRAIL pathway, a sequentially functionalized graphene-based nanostructure was developed as a new cellular protease-mediated programmed co-delivery system, integrating TRAIL protein and the intracellular-functioning small molecule doxorubicin (DOX). The nanocarrier was composed of the graphene oxide (GO) nanosheet, bearing DOX due to a strong supramolecular π–π stacking interaction; next, a heterobifunctional polyethylene glycol (PEG) linker was applied to connect the nanosheet with the cleavable peptide, which can be specifically recognized by furin, a protease highly expressed on the cell membrane and Golgi complex of cancer cells. Finally, TRAIL was conjugated to the sulfhydryl groups of the peptide using an amine-to-sulfhydryl crosslinker [29]. The working mechanism of the obtained TRAIL/DOX-fGO nanostructure involved both passive accumulation in the tumor due to the enhanced permeability and retention (EPR) effect and active TRAIL-directed tumor targeting, followed by the extracellular release of TRAIL after plasma membrane-located furin digestion of the peptide linker. Previously it was shown that DOX synergizes with TRAIL to overcome the resistance of human lung adenocarcinoma A549 cells [30]. In line with that, the GO nanosheet carrying DOX internalized into the cells and accumulated in the nuclei to produce DNA damage-mediated cytotoxicity, which synergized with TRAIL signaling for antitumor activity in the xenograft A549 tumor-bearing nude mice. TRAIL/DOX-fGO can serve as a model 2D nanocarrier with programmed-release therapeutics capability [29]. Further, molecular docking and molecular dynamics simulations have shown that the adsorption of TRAIL on graphene nanoflakes as a potential cargo is feasible and could lead to the recruitment of the receptors DR4 and DR5 with enhanced efficacy toward the targeted cancer cell. Thus, solid nanoparticles based on graphene flakes are perspective for multiple spatial drug deliveries, including TRAIL protein, due to their form and size [31].

Despite most nanocarriers associated with the TRAIL pathway mainly targeting DR4 and DR5 receptors by TRAIL, or DR5 selectively by receptor-specific agonists, in a few works, specific DR4-targeting nanocarriers were developed. Another GO-based nanocarrier was modified with the death receptor 4 (DR4) antibody and the siRNA for AKT (protein kinase B), a significant downstream protein kinase molecule of the PI3K signaling pathway, playing a vital role in the cell survival and growth [32]. The surface of graphene oxide (GO) was functionalized with carboxyl groups to link the DR4 antibody through a covalent bond (GO+A). Additionally, it was decorated with 1-pyrenemethylamine hydrochloride (PyNH_2_) with π–π interaction so that it could further attract the negatively charged AKT siRNA on the GO by electrostatic attraction (GO+A+S). The obtained nanocarrier targeted cancer cells overexpressing DR4 on their plasma membrane and induced the clustering of DR4s to activate the apoptosis signaling pathway. AKT siRNA disturbed the formation of AKT and alleviated the suppression of caspase-8, jointly contributing to cancer cell death. As a result, GO+A+S synergistically inhibited the growth of xenograft MCF-7 tumors in a mouse model with an enhanced therapeutic effect [32]. Therefore, a strategy for DR4 aggregation-mediated apoptosis may also be promising for cancer treatment.

Another example of graphene-based material for TRAIL pathway targeting is graphene quantum dots (GQDs), composed of sp2-hybridized carbon atoms, which are obtained by cutting a graphene monolayer into 2–20 nm disks [33]. In recent work, TRAIL fused with a crystalline bacterial cell surface layer protein (S-TRAIL) was conjugated non-covalently to graphene quantum dots (GQDs) (Figure 2). In this nanohybrid system, the self-assembly of the S-layer protein improved the stability and biological function of TRAIL, while the GQDs provided both a biocompatible surface for self-assembly of the fusion proteins and enabled the tracking of the cargo due to its unique optical properties. The anticancer efficacy of the nanohybrid system rated about 80% apoptosis in TRAIL-resistant HT-29 human colon carcinoma cells. Such tracking of TRAIL by taking advantage of GQD fluorescence highlights the potential of GQDs as a theranostic platform targeting TRAIL death receptors [34].

#### 2.2.2. Carbon Nanotubes (CNTs)

Carbon nanotubes (CNTs) possess exceptional physical properties, such as metallic conductivity, chemical and thermal stability, mechanical resistance, elasticity, flexibility, and the ability for chemical functionalization [35]. This makes CNTs feasible for nanovector engineering for medical applications. The same research group that developed TRAIL-functionalized iron oxide nanovectors [24,25,26] has vectorized TRAIL by single-walled carbon nanotubes (SWCNTs) via the non-covalent 1-pyrenebutanoic acid N-hydrosuccinimide ester (PSE), intending to increase TRAIL valency and enhance apoptosis by mimicking membrane TRAIL. The obtained TRAIL-functionalized nanovectors (NPTs) improved TRAIL affinity to DR5 and increased its pro-apoptotic potential by nearly 20-fold in human tumor cell lines of different origins, remaining highly selective for TRAIL signaling [36]. This proved that TRAIL nanovector derivatives based on SWCNT might be useful in cancer therapy. Importantly, it was shown that when docked to DR5, NPT carrying TRAIL homotrimer led to a more stable complex than TRAIL monomer-based NPT [37], supporting the other works which emphasized the importance of sterical orientation and the spatial arrangement of the TRAIL molecules for effective clustering of the death receptors and apoptosis induction [38]. Additionally, the authors confirmed the temperature dependence of the binding affinities of the soluble TRAIL and its nanovectorized form to its cognate receptors DR4 and DR5. At 37 °C (close to physiological conditions), the rank-ordered ligand-receptor affinities were at their maximum and strongly differed in the sequence: TRAILDR4 < NPTDR4 < TRAILDR5 < NPTDR5 [39]. This proves that the nanovectorization of TRAIL enhances its binding to both DR4 and DR5 receptors, with DR5 being more important for TRAIL signaling.

### 2.3. Boron Nitride Nanotubes

A formulation similar to the abovementioned was obtained with boron nitride nanotubes (BNNTs), which are analogous to SWCNT and have attracted considerable attention due to their chemical inertness, piezoelectric property, biocompatibility, and thermal and mechanical stability [40]. Herein, the TRAIL protein was anchored to 1-pyrenebutyric acid N-hydroxysuccinimide ester-functionalized BNNTs. The resulting nanotubes were mixed with methoxy-poly(ethylene glycol)-1,2-distearoyl-sn-glycero-3-phosphoethanolamine-N-conjugates to obtain nanoparticle TRAIL (NPT) with good dispersion in aqueous solution. The rank-ordered binding affinities of TRAIL and NPT ligands to DR4 and DR5 were different in the sequence TRAILDR4 < NPTDR4 < TRAILDR5 < NPTDR5 [41]. These data were fully consistent with the affinity values obtained previously for TRAIL formulation with SWCNT by the same authors [39], highlighting the universal character of TRAIL death receptor activation.

### 2.4. Periodic Mesoporous Organosilica (PMO) Nanoparticles

An innovative TRAIL pathway-targeting drug delivery system was recently fabricated from periodic mesoporous organosilica (PMO) nanoparticles, the optimized biocompatible inorganic–organic hybrids for drug delivery with large surface areas and tunable pore and particle sizes [42]. TRAIL, fused with the C-terminus fiber shaft of human adenovirus type 5 (HA5ST) and modified with cysteine, was conjugated on the maleimide-modified PMO surface, and the fabricated PMO-hT was loaded with doxorubicin in the PMO channels to obtain DOX@PMO-hT. This drug delivery system showed a synergistic effect due to the induction of tumor death through the TRAIL apoptotic pathway and immunogenic cell death in tumors, together with activating immune cells through the immunomodulatory functions of PMOs [43]. Such a drug delivery nanovehicle, which combines the properties of inorganic and organic materials and synergistically modulates immunity, may gain further development for anticancer treatment.

The inorganic nanoparticles modified with TRAIL pathway-targeting ligands, which are described above, are summarized in Table 1.

## 3. Lipid-Based Systems for Nanodelivery of TRAIL Pathway-Targeting Ligands

### 3.1. Liposomes

The liposome-based nanomedicines mainly consist of one or two therapeutic agents encapsulated in a spherical bilayer. They are used in clinics to deliver a wide range of therapeutic agents, including small molecules, peptides, and nucleic acids [44]. One of the most studied TRAIL liposomal formulations is LUVDOX-TRAIL, doxorubicin-loaded large unilamellar vesicles (LUV) with soluble TRAIL anchored on their surface. It comprises a sequel of the liposome-bound TRAIL development widely studied in various tumor models [45,46,47,48,49]. In one of the latest works on the topic, these double-edged lipid nanoparticles demonstrated the concerted action of the liposomal DOX and TRAIL in a fibrosarcoma HT1080-xenograft model without systemic toxicity in vivo [50].

Another liposomal formulation is surface-conjugated with TRAIL and the adhesion receptor E-selectin (ES/TRAIL), aiming to target and kill circulating tumor cells (CTCs). This is a continuation of long-term research, where TRAIL has been conjugated on the surface of nanoscale liposomes along with the adhesion receptor E-selectin (ES), which can recognize and bind to leukocytes. The efficacy of TRAIL/ES liposomes was previously evaluated by their ability to neutralize CTCs in the bloodstream [51], to prevent the spontaneous formation of metastatic tumors in an orthotopic xenograft model of prostate cancer [52], and to reduce metastasis following tumor resection in an aggressive triple-negative breast cancer (TNBC) mouse model [53]. In the most recent studies, ES/TRAIL liposomes demonstrated a potent synergistic apoptotic effect in prostate cancer cells with the liposomal form of piperlongumine [54] and killed CTCs and CTC clusters in blood samples from prostate cancer patients under flow conditions [55], asserting that liposomal TRAIL can be a promising adjuvant therapy. 

Additionally, a similar liposomal formulation developed by the same research group contained thiolated TRAIL together with anti-CD57, a monoclonal antibody against human natural killer cells, both functionalized on the liposome surface via maleimide-thiol chemistry. The TRAIL- and anti-CD57-functionalized liposomes were further conjugated with CD57-expressing NK cells, and the obtained “super” NK cells induced significant apoptosis in cancer cells of different origins [56]. In continuation of this work, liposomes functionalized with TRAIL and the antibody against the NK1.1 antigen expressed on murine NK cells were tethered onto the surface of NK cells, thus creating “super” NK cells, which prevented the lymphatic spread of a subcutaneous colon cancer tumor in mice [57]. One combined liposomal delivery system with tumor microenvironment responsiveness (TRAIL-[Lip-PTX]C18-TR) was aimed to co-deliver TRAIL and paclitaxel (PTX) for melanoma treatment. To obtain the multifunctional liposomal delivery system, PTX was loaded inside the liposomes, positively charged TRAIL was attached electrostatically to the negatively charged liposome surface, and finally, a stearyl chain (C18) fused with histidine-rich cell-penetrating peptide (CPP) named TR was attached to the surface of the liposomes by electrostatic and hydrophobic interactions. Despite PTX having been shown earlier to synergize with TRAIL in some other types of cancer [58], it has poor solubility in aqueous solutions and a low therapeutic index. The TR peptide not only selectively recognized cancer cells expressing integrin αvβ3 but also displayed pH-triggered cell penetration activity in the tumor microenvironment through charge conversion and consequent structure shift, which could be followed by pH-triggered cargo release from the liposomes with attached TR. As a result, the obtained TRAIL-[Lip-PTX]C18-TR displayed better drug release profile, stability, and circulation time in vivo, and improved tumor suppression effect [59]. Interestingly, neither TRAIL nor PTX has previously been shown to be effective against melanoma.

The next liposomal TRAIL preparation has been developed for a non-antitumor application, namely, to target the activated hepatic stellate cells (aHSCs) for the treatment of liver fibrosis. Apoptosis of aHSCs is an important mechanism for liver fibrosis recovery. NK cells can attenuate liver fibrosis by promoting apoptosis of aHSCs through TRAIL/DR5 pathway, revealing the role of TRAIL in liver fibrosis therapy [60]. To develop a new method to treat liver fibrosis, the TRAIL protein was chosen as cargo in a liposomal drug nanocarrier designated pPB-SSL. The sterically stabilized liposomes in pPB-SSL were specifically targeted for aHSCs by the cyclic peptide C*SRNLIDC* (pPB) with a specific affinity for platelet-derived growth factor receptor-β (PDGFR-β), which is predominantly expressed on the surface of aHSCs. The obtained pPB-SSL-TRAIL notably increased the apoptosis of aHSCs and remarkably alleviated hepatic fibrosis in mice compared to free TRAIL and SSL-TRAIL (TRAIL capsulated within unmodified liposome). As pPB-SSL can deliver TRAIL primarily into liver tissue with a significantly decreased drug retention in other types of organs, it may be promising for the therapy of liver fibrosis [61].

### 3.2. Solid Lipid Nanoparticles

Solid lipid nanoparticles (SLNs) offer several advantages, such as improved drug stability, high entrapment efficiency, and biocompatibility. Some SLNs have been applied for the treatment of TNBC by targeting the DR5 receptor, which is overexpressed on the surface of TNBC cells. For example, DR5-DAPT-SLNs is the SLN formulation surface-modified with the DR5 antibody and loaded with N-[N-(3,5-difluorophenacetyl)-L-alanyl]-Sphenylglycine t-butyl ester (DAPT), a potent γ-secretase inhibitor (GSI). DAPT targets Notch signaling, which is crucial for the maintenance of tumor-initiating breast cancer stem cells (BCSCs) in recurrent TNBC. However, it has a number of clinical limitations, such as low cellular bioavailability and off-target effects on healthy tissues, which can be overcome by loading into nanocarriers. The DR5-DAPT-SLNs enhanced cellular uptake and cytotoxicity compared to non-targeted SLNs and showed greater tumor regression in vivo compared to DAPT-SLNs and DAPT alone [62]. This lipid-based nanodelivery system can both selectively target cancer cells and potentiate the anticancer efficacy of DAPT against TNBC cells.

In a recent work, another DR5-targeted antibody, conatumumab (C), decorated the nanostructured lipid carriers (NLC) with co-encapsulated irinotecan prodrug (I-p) and a natural bioflavonoid quercetin NLC (Q). The NLC were composed of a binary mixture of solid lipid and a spatially different liquid lipid as the carrier. The obtained C I-p/Q NLC showed higher cytotoxicity than non-decorated NLC, single drug-loaded NLC, and free drugs on colorectal cancer (CRC) cells in vitro and in xenograft mouse models of CRC in vivo, asserting that the developed combination nano-system is a promising platform for CRC therapy [63].

The lipid-based systems for nanodelivery of TRAIL pathway-targeting ligands, which are described above, are summarized in Table 2.

## 4. Polymer-Based Systems for Nanodelivery of TRAIL Pathway-Targeting Ligands

### 4.1. Natural Polysaccharides-Based Drug Delivery Systems

Polymeric nanodelivery systems obtain vast variability due to polymer variety and tunability, which allows them to be used in versatile biomedical applications [64]. Natural polysaccharides, such as chitosan and alginates, are bioactive polymers, popular in the fabrication of nanodelivery systems for their controlled drug delivery, enhanced safety, biocompatibility, bioavailability, better retention time, lower toxicity, and enhanced permeability. Alginate (ALG) is a biocompatible polysaccharide extracted from brown seaweeds, composed by a sequence of two (1-4)-linked α-L-guluronate and β-D-mannuronate monomers, and obtains very low toxicity, which makes it one of the biopolymers with the widest biomedical applicability, particularly in drug delivery systems [65]. In an early work, a nanocomposite with a porous DOX-loaded CaCO_3_ core electrostatically functionalized with TRAIL, and an alginate multilayer shell was fabricated by a layer-by-layer (LbL)-assembly technique. The obtained TRAIL/ALG-CaCO_3_ nanocomposites displayed notable anticancer activity in vitro [66].

Chitosan is a well-known natural linear hetero-polysaccharide extracted from crabs and shrimp comprised of cationic residues β-1,4- linked 2-amino-2-deoxy-glucopyranose and 2-acetamido-2-deoxy-βD-glucopyranose. Due to its low immunogenicity, biocompatibility, non-toxicity, antibacterial activity, biodegradability, film forming, and emulsifying characteristics, chitosan is approved by the FDA for pharmaceutical and clinical applications, in particular, the delivery of bioactive compounds. The amino functional groups of chitosan are suitable for biochemical adjustments and electrostatic interactions [65]. To target the TRAIL receptor pathway, the glycol chitosan nanoparticles (GCS) were loaded with anti-human DR5 single-chain antibody fragments (aDR5 scFv) by ion gelation. The obtained GCS-aDR5 scFv targeted the tumors much more efficiently than healthy organs, whereas the plain nanoparticles had no such targeting ability. Additionally, the GCS-aDR5 scFv effectively inhibited tumor growth of the xenograft hepatocellular H22 cells and exhibited high stability [67].

The chitosan-based nanotherapeutics were also applied in an alternative approach that utilized the TRAIL-related pathway in an opposite way. Since TRAIL elevated expression in tumor cells can distinguish them from healthy cells, the TRAIL protein itself was targeted by the hybrid nanoparticles composed of a chitosan polymer core with encapsulated oxaliplatin, which was further coated by the outer negative charged lipid layer to prevent unfavorable aggregation. 1,2-distearoyl-sn-glycero-3-phosphoethanolamine-polyethylene glycol-3400 maleimide (DSPE-PEG3400 mal) was added to enable surface covalent conjugation with the thiolated anti-TRAIL(CD-253) antibody. The obtained oxaliplatin immunohybrid nanoparticles (OIHNPs) effectively delivered the drug to the tumor sites, showing sustained release and effective reduction of tumor volume in xenograft HT-29 tumor models in vivo. This may be a promising nanosized formulation with an alternative targeting of a key component of the TRAIL pathway for tumor-targeted therapy [68].

Another combined delivery system for the DR5-targeting antibody (AMG655) was composed of a hydrogel-based chitosan/alginate nanoparticle formulation with encapsulated meso-Tetra(N-methyl-4-pyridyl) porphine tetratosylate (TMP), a hydrophilic photosensitizer that can be used in photodynamic therapy (PDT) to induce cell death through the generation of reactive oxygen species in targeted tumor cells. AMG655 was chemically conjugated to the nanoparticles by carbodiimide chemistry via available amino and carboxyl groups to exposed reciprocal carboxyl and amino groups on the alginate and chitosan polymers, respectively. As a result, antibody-conjugated chitosan/alginate nanoparticles (DR5 TMP NP) significantly enhanced the therapeutic effectiveness of entrapped TMP, providing a strategy for targeted site-specific drug delivery for antitumor treatment. The DR5-targeted nanoparticles induced activation of caspase 8, whereas a comparable amount of free DR5 antibody was unable to activate the receptor [69].

### 4.2. Poly(lactic acid) (PLA) and Poly (lactic-co-glycolic acid) (PLGA)-Based Drug Delivery Systems

Poly(lactic acid) (PLA) and poly (lactic-co-glycolic acid) (PLGA) are among the most successfully used synthetic polymers for biomedical applications because their hydrolysis leads to the metabolite monomers lactic acid and glycolic acid [70]. Due to excellent biodegradability and biocompatibility, various modifications of PLA- and PLGA-based polymers have been widely used to develop therapeutic nanocarriers.

The anti-DR5 monoclonal antibody AMG 655 (conatumumab) and several other first-generation DR5-targeting antibodies failed in clinical trials due to a low therapeutic effect [71]. One of the probable reasons is the steric inconsistency of dimeric antibodies with trimeric DR5 receptors, requiring crosslinking of DR5 for efficient receptor clustering and apoptosis induction. Alternatively, attempts have been made to increase the valency of antibodies by attaching them to the surface of nanoparticles. The same research group that fabricated AMG655-conjugated chitosan/alginate nanoparticles [69] also demonstrated that efficient DR5 receptor clustering could be achieved through conjugation of AMG 655 to PLGA nanoparticles using carbodiimide chemistry, leading to the activation of apoptosis in colorectal models [72,73]. The conjugation of AMG 655 to the PLGA nanoparticles resulted in a similar activation of caspase 8, as with the chitosan/alginate nanoparticles, suggesting that these particles are able to act by a similar mechanism. In the novel formulation, named αDΡ5-NPs, an NHS-modified PLGA-PEG was incorporated in the polymer blend, which enables one-step conjugation to AMG 655 after preparation of the nanoparticles, thus improving the fabrication scalability. Compared to free AMG 655, the αDΡ5 NPs showed an enhanced ability to induce apoptosis in pancreatic cancer cells. Additionally, αDΡ5-NPs were loaded with the DNA topoisomerase 1 inhibitor camptothecin (CPT) (CPT DR5 NP), leading to synergistically enhanced efficacy by causing downregulation of FLIP(L) and FLIP(S). CPT-loaded αDΡ5-NPs had more potent activity in vitro than either free CPT, nude CPT-loaded NPs, or non-drug-loaded αDΡ5-NPs, thus sensitizing previously resistant tumors to DR5-targeted therapy [74]. Antibody-conjugated nanoparticles are an attractive alternative to antibody-drug conjugates as they not only elicit increased avidity due to multivalent effects but also offer reduced drug toxicity [75]. Thereby, this is a promising direction for DR5-targeted nanodelivery development.

A similar approach utilized ultrasound contrast agents consisting of polymeric microbubbles (MBs) as a drug delivery vehicle. The advantage of using the MBs as a drug delivery platform is their ability to be visualized by the inertial cavitation of the gas core upon focusing the ultrasound beam in the tumor. These stabilized MBs are disrupted into the drug-loaded fragments, nanoshards, only in response to ultrasound focused at the tumor site. The resulting nanoshards are small enough (up to 400 nm) to exit the leaky tumor vasculature and provide sustained release of the encapsulated drug at the targeted tumor site. The native shell consisting of PLA and PEG was inserted into the shell to decrease the immunogenicity. The MB surface was functionalized by TRAIL covalently via maleimide-thiol chemistry. Additionally, the functionalized MBs were designed to co-encapsulate doxorubicin (DOX) [76]. The study of the acoustic responsiveness of the developed TRAIL-ligated and DOX-loaded MBs showed that they were capable of interacting with ultrasound under physiological conditions [77]. Importantly, the further evaluation of biological activity in TRAIL-sensitive and TRAIL-resistant breast cancer cells showed that for all shell types, nanoshards had a greater effect compared with full-size MBs, reflecting the greater surface area and the larger number of particles that ultrasound generates [78]. This indicates that the MBs can direct drug-loaded nanoshards into the tumor in response to ultrasound focused at the site, shielding healthy tissues from the toxicity of DOX while also increasing the potency and efficiency of the TRAIL antitumor treatment.

In support of the aforementioned data, recent studies of the effect of a nanoparticle’s shape on its function have shown that the traditional spherical shape may not be optimal for drug delivery compared with the high length-to-width ratio of anisotropic nanoparticles. For example, rod-shaped nanoparticles may be less recognized and taken up by phagocytes and obtain enhanced margination, extravasation, and tissue penetration. Additionally, elongated nanoparticles have a large contact surface area, which allows for more efficient exposure of ligands compared to spherical nanoparticles, whose high curvature can prevent multivalent interactions with target receptors on the cell surface [79]. Matching the above, it was shown that PLGA is also capable of forming elongated nanofibers in addition to globular nanostructures. A research group that recently designed the highly stable and active recombinant sTRAIL homotrimer via genetic engineering using the adenovirus knobless fiber motif [80] has advanced their formulation by encapsulating TRAIL into PLGA nanofibers by coaxial electrospinning to obtain controlled drug release. TRAIL maintained its biological activity during the electrospinning process and effectively inhibited the growth of breast cancer tumor cells as a part of PLGA/sTRAIL nanofiber mats. Surprisingly, TRAIL encapsulated in nanofibers inhibited xenograft breast tumor growth more efficiently upon subcutaneous implantation compared to intraperitoneal injection. Therefore, the implantation of TRAIL-encapsulated nanofibers is a conceptually viable antitumor therapeutic strategy [81].

In addition to traditional TRAIL death receptor targeting, PLGA nanoparticles have been used to affect the TRAIL-dependent pathway in an alternative way. The authors previously showed that TRAIL represents an important target in a brain impacted by Alzheimer’s disease (AD) [82]. In a recent study, the TRAIL-neutralizing monoclonal antibody was adsorbed onto the surface of polymeric (PLGA, NANO-A) and lipidic (NLC, NANO-B) nanoparticles to reach the brain through the nose-to-brain (N2B) route. Both NANO complexes were able to reach the brain and localize in the hippocampal areas of AD mice models in substantial amounts at concentrations significantly higher compared to the free form of the anti-TRAIL antibody. This represents an advantageous tool for the non-invasive treatment of neurodegenerative disorders [83].

Another set of studies has applied a flexible approach by combining the advantages of PLGA nanoparticles with the unique properties of the cell membrane. For example, TRAIL-functionalized drug-loaded NPs coated with an umbilical vein endothelial cell (UVEC) membrane (TU-NPs) were fabricated for antirheumatic drug delivery by targeting the activated M1 macrophages in the inflammatory sites of rheumatoid arthritis (RA). Inflammatory M1 macrophages play a crucial role in the progression of RA, and upregulated DR5 receptors make it susceptible to TRAIL-induced apoptosis. To prepare RA joint targeting TU-NPs, the synthesized PLGA cores were loaded with hydroxychloroquine (HCQ), a first-line antirheumatic drug with low adverse effects, and further coated by the plasma membrane of UVECs, stably expressing TRAIL. The therapeutic efficacy of TU-NPs was demonstrated in the collagen-induced arthritis (CIA) mouse model [84]. 

In another study, PLGA was employed as the core, which was preloaded with the chemotherapeutic drug paclitaxel (PTX) and camouflaged with a neutrophil membrane (NM). Further, TRAIL was introduced into the hybrid system (TNM-PN) to bind to tumor cells for the promotion of cellular internalization and further apoptosis induction. TNM-PN exerted significant cytotoxicity to tumor cells by TRAIL-mediated endocytosis and strong adhesion to inflamed endothelial cells in vitro. Prolonged blood circulation and preferential tumor accumulation of TNM-PN resulted in significant tumor inhibition and enhanced animal survival rates. Thus, PLGA coating by NM amplified cellular internalization and boosted antitumor efficacy [85]. 

At last, a novel recently reported biosynthetic TRAIL-sensitizing nanoplatform, CPT MV, combined the advantages of polymeric nanoparticles and membrane vesicles. CPT MV nanospheres consisted of a PLGA core loaded with photosensitizer chlorin e6 (CP NPs) and coated with a thin film of TRAIL-expressing cell membrane vehicles (TRAIL MV), which were obtained from lentiviral murine TRAIL-expressing genetically engineered CHO-S cells. This designed photo-triggered nanoplatform could produce ROS in the targeted cells upon laser irradiation to improve DR5 expression and trigger cytochrome c release from mitochondria. The strengthened TRAIL functionality resulted in strong cell apoptosis and suppression of xenograft murine hepatoma hepa 1-6 tumors, along with high biosafety. Such a photosensitizer-containing nanomedicine camouflaged with a TRAIL-expressing cell membrane is a perspective candidate for treating TRAIL-resistant tumors by improving the therapeutic outcomes of TRAIL-based cancer treatment [86].

### 4.3. Nanogels

Chemical crosslinking or physical self-assembly of specific polymers allows for the fabrication of nanogel-based multifunctional drug delivery nanoplatforms, a tremendously promising system of drug delivery with the ability to encapsulate a wide range of hydrophilic or hydrophobic therapeutics. Nanogels are characterized by a specific surface area and inner space, providing controlled and stimuli-responsive drug release and accumulation in various microenvironments [87]. One of the most recent developments is the gel-like mPEGylated coacervate (mPEG-Coa) delivery platform, preserving the bioactivity of cargo TRAIL for colon cancer treatment [88]. A gel-like colloidal droplet coacervate (Coa) structure was fabricated by the electrostatic interaction of the polycation poly(ethylene arginyl aspartate diglyceride) (PEAD) with polyanionic heparin followed by self-assembly in aqueous conditions. Additionally, methoxy-poly(ethylene glycol) (mPEG) was conjugated on the PEAD cation backbone to augment the colloidal stability of Coa. The TRAIL protein, which was incorporated into the obtained gel-like mPEG-Coa droplets, was protected against protease activity by providing a long-term (14 days) sustained release and suppressed colon cancer cell recurrence as compared with free TRAIL. This gel-like mPEG-Coa-based TRAIL delivery system is suggested as a novel protein drug delivery platform to suppress cancer recurrence after resection or chemotherapy by the prolonged supply of exogenous TRAIL [88].

Another recent study aimed to develop a nanogel-based therapy that eliminated the extracellular matrix barrier to increase TRAIL delivery into tumors and also blocked antiapoptotic mechanisms to overcome TRAIL resistance in pancreatic ductal adenocarcinoma (PDAC). A nanogel modified with phage display-identified desmoplastic tumor stroma-targeting peptides was constructed to co-deliver nitric oxide (NO) and TRAIL to PDAC (Figure 3). The fabricated nanoparticles, termed TRAIL-NO@Nanogel, consisted of a silk fibroin (SF) hydrogel core loaded with TRAIL and a shell composed of lipids and PLGA loaded with a synthetic NO donor dinitrosyl iron complex (DNIC; Fe(μ-SEt)_2_(NO)_4_). The delivery of NO to the PDAC tumor stroma resulted in the reprogramming of activated pancreatic stellate cells, alleviation of tumor desmoplasia, and suppression of antiapoptotic Bcl-2 protein expression, resulting in facilitated tumor penetration by TRAIL and enhanced TRAIL antitumor efficacy in three-dimensional spheroid cultures in vitro and in orthotopic PDAC models in vivo. Thus, the co-delivery of TRAIL and NO by a stroma-targeted nanogel that remodels the fibrotic tumor microenvironment and suppresses tumor growth is promising for PDAC treatment [89].

Since TRAIL is a multifunctional cytokine executing various functions in the immune system, expectedly, non-antitumor TRAIL applications are mostly devoted to its immune-related activity, particularly for the treatment of inflammatory diseases caused by bacterial infections. For this, TRAIL was encapsulated in a bactericidal crosslinked cationic poly(L-lysine)-block-poly(L-threonine) (PLL-b-PLT) co-polypeptide nanogel to target excessively activated macrophages upon *Klebsiella pneumoniae* infection [90]. The mechanism included the targeting of the outer membranes of bacteria made of lipopolysaccharides (LPS) by nanogels through electrostatic interactions. Next, the LPS-overactivated macrophages phagocytized the complex and induced apoptosis due to the release of TRAIL. Intraperitoneal injections of TRAIL-encapsulated nanogel protected mice against *K. pneumoniae*-induced sepsis, significantly prolonged survival in septic mice, and reduced bacterial numbers in circulation. Therefore, TRAIL-encapsulated nanogel is stated to be a promising agent for treating bacterial infections [90].

### 4.4. Micellar Nanoparticles

Linear amphiphilic block copolymers are able to form nanoscale core–shell micellar particles via a self-assembly process in an aqueous environment, with a hydrophobic block forming the core to minimize aqueous exposure and a hydrophilic block forming the shell to stabilize the core [91]. Polymeric micellar formulations benefit in improved drug solubility, selective tumor targeting, reduced adverse effects, and prolonged circulation time cells [92]. For example, the polymeric self-assembled micellar formulation was developed from biodegradable amphiphilic copolymers, monomethoxyl poly(ethylene glycol)–b-poly(DL-lactide) (mPEG-PLA), and COOH-PEG-PLA via a nanoprecipitation method. The micelles were loaded with a very low water-soluble chemotherapy drug, cabazitaxel (CTX), and the surface was modified with the TRAIL protein via carbodiimide chemistry. The obtained micellar nanoparticles, TRAIL-M-CTX, showed an improved anticancer efficacy than the individual CTX and TRAIL protein, along with synergistic effects against TRAIL-resistant cells [93]. 

Another co-delivery of polymeric nanoparticles was composed of poly (e-caprolactone) (PCL) as a hydrophobic segment and PEG as a hydrophilic segment to avoid aggregation and clearance from the reticular-endothelial system. The obtained biodegradable poly (e-caprolactone)-poly (ethylene glycol)-poly (e-caprolactone) (PCEC) triblock copolymer nanoparticles were loaded with curcumin (Cur) and coated with TRAIL protein via electrostatic interactions. The resulting TRAIL-CurNPs improved the therapeutic effect of the drugs by prolonging their circulation time, enhancing the water dispersion of curcumin, and improving active and passive tumor targeting. TRAIL-Cur-NPs demonstrated enhanced cellular uptake and apoptosis induction, as well as a superior therapeutic effect on xenograft HCT116 colon cancer cells in vivo compared with corresponding free drugs, without obvious toxicity [94].

To develop complex micellar polymeric nanocomposites targeting the TRAIL pathway, a combinational strategy was applied. For this, iron oxide nanoparticles (IO) were encapsulated into the core of micellar nanoparticles, composed of self-assembled PEGylated cationic amphiphilic polymers (Alkyl-PEI2k-PEG2k), together with a new photothermal agent, metalla-aromatics complex Ph556. Next, these hybrid micellar nanoparticles (IO+Ph556@NPs, abbreviated as IPN) adsorbed negatively charged TRAIL ligands via electrostatic interactions, therefore obtaining TRAIL nanocomposites (IO+Ph556@NPs@TRAIL, abbreviated as IPN@TRAIL) [95]. This formulation optimized TRAIL biodistribution and tumor accumulation, improving TRAIL anticancer activity towards resistant A549 cancer cells due to nanovectorization. However, and more importantly, employing photothermal therapy (PTT) by photoabsorber Ph556 converting light to heat for tumor thermal ablation caused a synergistic effect between IPN nanoparticles and wrapped TRAIL protein by upregulating the expression of the TRAIL death receptors DR4 and DR5, resulting in the synergistic effects between PTT and TRAIL therapy. Usually, chemotherapeutic agents are used to upregulate the expression of death receptors DR4 and DR5 in cancer cells [96]. The remodeling of the death receptor expression by PTT is stated as safer and more controllable compared to chemotherapy [95].

Recently, our group fabricated nanoparticles based on the biocompatible polymer amphiphilic poly(N-vinylpyrrolidone) (Amph-PVP), which is capable of self-aggregating in aqueous solutions with the formation of micellar nanoscaled structures. The Amph-PVP nanoparticles were obtained from a 1:1 mixture of unmodified and maleimide-modified polymeric chains and further conjugated with the DR5-specific TRAIL variant DR5-B [97], modified with the N-terminal cysteine residue for covalent reaction by maleimide-thiol chemistry. The obtained P-DR5-B nanoparticles enhanced the cytotoxicity of free DR5-B proteins in vitro in monolayer cell cultures and multicellular tumor spheroids of colon carcinomas HCT116 and HT29 and breast adenocarcinoma MCF-7 cells without cytotoxicity to normal cells. Taking into account the Amph-PVP ability to be loaded with a wide range of low-molecular-weight antitumor chemotherapeutics into hydrophobic core and the feasibility of conjugation with hydrophilic therapeutic molecules, it can be further developed into a versatile system for targeted drug delivery to tumor cells [98].

The polymer-based systems for nanodelivery of TRAIL pathway-targeting ligands, which are described above, are summarized in Table 3.

## 5. Protein-Based Systems for Nanodelivery of TRAIL Pathway-Targeting Ligands

### 5.1. Protein Nanoparticles

When discussing peptide-based nanovehicles, it is necessary to mention the well-established protein nanoparticles, in particular those based on albumin. Despite serum albumin nanoparticles having already been well-established for drug delivery, they are still widely studied to expand their range of applications in therapy and diagnostics [99]. For example, a newly developed bovine serum albumin (BSA) based nanodelivery system relies on the unique characteristics of the tumor microenvironment, one of which is the acidified extracellular pH. In the reported pH-responsive nanoparticles, BSA NPs were used as a core for loading gambogic acid (GA). Subsequently, the BSA NPs were adsorbed by polyethylenimine and then shielded with carboxymethyl chitosan-folate (CMCS-FA) as the outer shell for encapsulating TRAIL protein by the layer-by-layer (LbL) assembly technique, forming the GA/TRAIL co-delivery BSA (GTB) NPs for tumor targeting and synergistic therapy. CMCS-FA is a sheddable shell material that is sensitive to tumor extracellular pH, and folate exposed on the surface of the nanocarriers facilitates endocytosis by the folate receptor. In normal tissues, the GTB NPs are negatively charged; however, in acidic tumor tissues, the shielding CMCS-FA is detached, allowing the release of TRAIL, which binds to the cell death receptor on the plasma membrane. The resulting positively charged complex promotes cellular internalization and escape from lysosomes, producing a rapid release of GA, which exerts the combined tumor therapy. GTB NPs displayed an improved antitumor efficacy in the MCF-7 xenograft model and reduced side effects when compared with GA solution [100]. The authors did not compare the antitumor effect of GTB NPs with free TRAIL; however, this could be valuable information for further development of the pH-responsive nanodelivery therapeutic strategy.

The same research group that fabricated TRAIL/ALG-CaCO_3_ nanocomposites [66] has further developed DOX-loaded BSA nanoparticles with encapsulated oppositely charged TRAIL and alginate by consecutive adsorption using the LbL technique. The assembled core/shell structure of the nanoparticles was internalized more easily with the cancer cells. The co-delivery of TRAIL together with DOX in the obtained TRAIL/ALG-DOX@BSA formulation exhibited a synergistic cytotoxic effect on H460 cells and on the DOX-resistant L929 cells [101].

The same group of authors has further constructed protein-based nanoparticles with TRAIL and a dipeptide diphenylalanine (FF) through the supramolecular self-assembly method. FF is the core recognition motif of the β-amyloid peptide associated with the neurodegenerative disorder of Alzheimer’s disease, which can be used as a versatile small molecular building block for the preparation of various nanostructures. As the sequence of TRAIL274-275 is also the FF fragment, it can interact with the free FF via π–π stackings and hydrogen bonds. Therefore, FF was employed in an assembly with TRAIL to form TRAIL-FF nanoparticles in the aqueous medium. These nanoparticles could gather around the membrane of MCF-7 cells due to the specific interaction between TRAIL and death receptors and showed distinct cytotoxic effects on the cancer cells MCF-7 and H460 [102].

Another promising self-assembling biopolymer system for drug delivery is temperature-sensitive elastin-like polypeptides (ELPs). Due to the amphiphilic structural motif of human tropoelastin, genetically encoded ELP can self-assemble into nanoparticles for drug delivery when the temperature is raised above the transition value. Recombinant RGD-TRAIL fused to the ELP chimeric polypeptide self-assembled at 37 °C under physiological conditions, resulting in ligand multimerization and 3-fold enhanced apoptosis-inducing capacity over RGD-TRAIL without apparent toxicity [103]. ELP-based nanodelivery retains the advantages of polymeric drug delivery systems, such as precise control over biopolymer composition and molecular weight, biodegradability, and lack of toxicity. Therefore, RGD-TRAIL-ELP is suggested as an efficient nanosized antitumor therapeutic strategy. Additionally, these works can be helpful for the preparation of other protein-based nanoparticles for biomedical applications.

### 5.2. Protein Cages

Protein cages are nanostructures composed of self-assembling protein subunits capable of forming hollow structures with well-defined geometric shapes and an empty interior space that can be filled with therapeutic agents for drug delivery. Protein cage nanoparticles are excellent multifunctional delivery nanoplatforms due to their highly symmetric and uniform architecture, multivalent nature, the ability for genetic and chemical manipulations, and high biocompatibility. While the defined interior spaces can be filled with size-constraint diagnostic or therapeutic reagents, the exterior surface can be modified with various types of molecules, including affinity tags, antibodies, fluorophores, carbohydrates, nucleic acids, and targeting peptides. Additionally, protein cage nanoparticles can be generated as chimeric proteins for a multifunctional approach [104].

Among such nanocages, the ferritin nanocage is a well-studied biological nanoparticle capable of entrapping a variety of low-molecular substances within its cage structure. Human ferritin heavy chains self-assemble to form a constant 24-subunit structure with a spherical cage-like architecture, which is employed in various medical applications, including targeted drug delivery [105]. A novel, developed biomimetic delivery platform presented a highly stable TRAIL homotrimer on the surface of naturally occurring ferritin heavy chain nanocages. A spacer was added between the human ferritin heavy chain (hFTH) nanocage and the TRAIL trimer complexes to improve the accessibility of the receptor-binding motif and its subsequent binding affinity for the receptor. The whole genetically fused TRAIL-presenting nanocage (TTPN) construct was expressed as a recombinant protein in *E. coli*. The N-termini-fused TRAIL molecules formed a trimer-like structure at the three-fold symmetry axis on the surface of the ferritin nanocage. Since the 24 monomeric ferritin subunits self-assemble into the cage structure, a total of eight native-like TRAIL homotrimers were displayed on the surface of the ferritin nanocages. Thus, TTPNs were designed to present the trimeric TRAIL complex in its rigid, symmetrical, optimal-binding conformation. As a result, TTPN demonstrated enhanced stability and improved pharmacokinetic characteristics and antitumor bioactivity compared with monomeric TRAIL in vivo [106]. However, TTPNs showed substantial therapeutic efficacy only in a TRAIL-sensitive tumor model. 

To overcome therapeutic efficiency limitations against TRAIL-resistant tumors, the authors exploited a TTPN that delivers not only TRAIL but also doxorubicin [107]. As ferritin nanocages are cellular iron storage proteins with intrinsic metal binding sites, DOX can be encapsulated into them utilizing divalent metal ions. Therefore, DOX was complexed with Cu2+ to create Cu-DOX, and this complex was loaded into the inner cavity of the ferritin nanocage, encountering 30 ± 6 DOX molecules per TTPN. The resulting complex DOX-TTPNs efficiently sensitized TRAIL-resistant tumor cells to TRAIL-mediated apoptosis in vitro by regulating levels of DR5 and anti- and pro-apoptotic proteins involved in extrinsic and intrinsic apoptosis pathways. DOX-TTPNs also successfully sensitized TRAIL-resistant xenograft tumors of the colorectal cancer cell line HT29 to the apoptotic effects of TRAIL in vivo, even at a very low dose of the incorporated DOX, underscoring the potential of this platform as an antitumor agent [107].

Another work reported a similar ferritin protein nanocage delivery platform in which TRAIL trimer formation was aided by ligating the triple helical domain of pulmonary surfactant-associated protein D to the TRAIL C-terminus. The obtained TRAIL-conjugated ferritin active trimer nanocage TRAIL-ATNC contained flexible linkers consisting of small amino acids (GSGGGSG) that formed a bridge between the C-terminal of TRAIL and the triple helix. TRAIL-ATNC showed enhanced cytotoxic activity compared to monomeric TRAIL [108]. Additionally, interleukin 4 receptor (IL4R)-binding peptide (IL4rP) (CRKRLDRNC) was ligated to the C-terminal end as a tumor-targeting peptide moiety. IL4rP, homologous to the sequence of IL4, was earlier shown to selectively target IL4 receptor (IL4R)-expressing tumors by specifically binding to IL4R, a biomarker of tumor cells [109]. The resulting TRAIL-ATNC^IL4rP^ formed nanosized 24-mer cages with an average diameter of 22.4 nm and showed enhanced antitumor efficacy in an orthotopic pancreatic cancer model [108]. This trimer delivery platform using the multi-display of tumor targeting IL4rPs and TRAIL on the surface ferritin protein nanocage is another nanodelivery system overcoming drawbacks of TRAIL, such as titration by the decoy receptors and rapid elimination from the circulation.

Similarly, proteinoid nanocapsules have been engineered for TRAIL delivery [110]. Proteinoids are non-toxic biodegradable polymers prepared by the thermal step-growth polymerization of amino acids. Due to biodegradability, non-immunogenicity, and low toxicity, proteinoids have been successfully applied for drug delivery [111]. In one work, TRAIL was covalently conjugated to the surface of the self-assembled proteinoid nanocapsules (NCs) based on D-arginine (R), glycine (G), and L-aspartic acid (D) building blocks to form TRAIL-P(RGD) NCs. The RGD sequence was intended to provide additional tumor targeting due to specific binding to the αvβ3 integrin, which is widely expressed on tumor cells and tumor vessels [112]. TRAIL-P(RGD) exhibited high cytotoxicity in human ovarian cancer cells while retaining the efficiency of free TRAIL [110]. Thus, proteinoid nanocapsules potentially offer increased stability for the TRAIL ligand; however, additional in vivo studies are required.

One of the most recent research studies [113] aimed to overcome TRAIL resistance, which may be caused by the activation of a survival signal via the epidermal growth factor (EGF)/epidermal growth factor receptor (EGFR) signaling pathway [114]. Unlike the aforementioned studies, which were focused mostly on a polyvalent display of multiple copies of TRAIL on the surface of protein cage nanoparticles, here, the polyvalent display of two different types of ligands was developed. To simultaneously display EGFR modulators in addition to TRAIL on the surface of a single nanoparticle, a lumazine synthase protein cage was applied. It is a multiple-ligand-displaying nanoplatform aimed to polyvalently expose both TRAIL and EGFR-binding affibody molecules (EGFRAfb). The lumazine synthase is originally isolated from the hyperthermophile Aquifex aeolicus (AaLS) and consists of 60 identical subunits that self-assemble to form a hollow icosahedral capsid architecture [115]. ST and SC are split proteins derived from the second immunoglobulin-like collagen adhesin domain found in *Streptococcus pyogenes* [116], which spontaneously form an irreversible isopeptide covalent bond upon recognition. AaLS was genetically engineered to present 60 STs on its surface (AaLS-ST), whereas human TRAIL and EGFR-binding affibody molecules (EGFRAfb) were genetically fused to SC protein to constructs SC-TRAIL and SC-EGFRAfb, respectively. Further, they were polyvalently displayed on the surface of the AaLS-ST using an ST/SC post-translational protein-ligation system to form AaLS/TRAIL/EGFRAfb. Each AaLS-based nanoparticle carried approximately 30 copies of TRAIL and EGFRAfb molecules in an equal ratio without altering the protein cage architecture [113]. The cytotoxic effect against EGFR-overexpressing cancer cells was improved upon dual display of TRAIL and EGFRAfb on the AaLS (AaLS/TRAIL/EGFRAfb) compared with that of AaLS/TRAIL both in vitro and in vivo. This implies that the obtained AaLS/TRAIL/EGFRAfb not only disrupted the EGF-mediated EGFR survival signaling pathway by blocking EGF/EGFR binding but also strongly activated both the extrinsic and intrinsic apoptotic pathways. Importantly, to achieve synergistic effects, TRAIL and EGFRAfb molecules should be displayed on the same protein cage nanoparticles and should be clustered together [113]. However, the authors mentioned that despite its effective tumor targeting, AaLS/TRAIL/EGFRAfb may have limited tumor penetration capability in vivo. Perhaps, this could be a direction for further development, for example, by applying cell-penetration peptides.

### 5.3. Virus-Based Scaffolds

In an effort to explore novel materials suitable for clinical use, another group aimed to repurpose the rod-shaped potato virus X (PVX) for TRAIL delivery [117]. PVX is a plant virus belonging to the Potexvirus group; its virion is assembled from 1270 identical 25 kDa coat protein units arranged orderly around its single-stranded RNA, resulting in a flexible elongated nanoparticle. PVX is biocompatible, biodegradable, and non-infectious to mammal cells. It can be obtained via farming with a high degree of reproducibility and monodispersity. Since PVX carries multiple surface-exposed residues, such as lysine and cysteine, it can be modified to load multiple functional modules [118]; additionally, hydrophobic drugs such as doxorubicin can also be loaded into grooves of the protein assembly via hydrophobic interactions [119]. Such engineerability and structural properties make PVX a perspective nanoplatform for drug delivery. To develop therapeutic TRAIL-loaded PVX, TRAIL with an N-terminal His-tag was non-covalently conjugated with nickel-coordinated nitrilotriacetic acid (NiNTA) modules displayed on the PVX surface, resulting in an elongated nanoparticle displaying up 490 therapeutic protein molecules. PVX-delivered TRAIL outperformed soluble TRAIL in activating caspase-mediated apoptosis and in delaying tumor growth in an athymic nude mouse model bearing human triple-negative breast cancer xenografts in vivo. This work highlighted the potential of plant virus nanotechnologies for targeting protein drug delivery in cancer treatment. The authors challenge future directions by incorporating TRAIL into PVX by genetic fusion techniques for streamlining manufacturing and mitigation of potential toxicity from the nickel ion used for the complexation of His-tagged TRAIL [117].

In addition to plant virus-based nanoplatforms, TRAIL-linked oncolytic adenovirus rAd5pz-zTRAIL-RFP-SΔ24E1a (A4), coupled to capsid protein IX (pIX) by a synthetic leucine zipper-like dimerization domain (zipper), was developed. Thus, previously developed A4 carried TRAIL on its surface and was able to target tumor cells [120]. To further improve the therapeutic potential of A4, the authors coated it with additional soluble TRAIL, fused with a leucine zipper-like dimerization domain (zipper). The obtained ZA4 enhanced infectivity and inhibited the proliferation of acute myeloid leukemia (AML) cells from cell lines and primary patient samples that expressed moderate levels of TRAIL-related receptors. ZA4 also showed enhanced antitumor activity in vivo compared with A4 and an unmodified adenoviral vector. Thus, the TRAIL-conjugated oncolytic virus zA4 might be a promising anticancer agent [121].

Encouraging results have been recently obtained by a biomimetic strategy for versatile and robust TRAIL delivery through the supramolecular construction of virus-mimetic nanocapsules (VMNs) [122]. VMNs were composed of amphiphilic dendritic peptides mimicking the viral protein component and structure, hydrophobic lipoic acid for driving self-assembly into the capsid-like nanostructures by disulfide bond cross-linking, and dynamic negatively charged decoration. Positively charged TRAIL protein was entrapped onto the obtained artificial capsids (AC) through electrostatic interactions with the negatively charged AC surface. Finally, the whole construction was electrostatically coated with artificial envelopes composed of positively charged chitosan oligosaccharides (COS) decorated with Arg-Gly-Asp (RGD) peptides, generating TRAIL-loaded virus-mimetic nanocapsules (P-VMNs). Importantly, the mixed-charged surface of P-VMNs can resist protein adsorption for prolonged blood circulation. At a lower pH (pH 6.5) corresponding to tumor extracellular acid conditions, P-MVNs showed higher antitumor activity than that at pH 7.4 due to the triggering of the negative-to-positive AC charge conversion and TRAIL release. Additionally, P-VMNs’ virus-inspired design and the RGD-modified envelope contributed to the deep tumor penetration and accumulation of TRAIL. This virus-mimicking delivery system successfully guided TRAIL to surmount sequential biological barriers and improve the in vivo efficacy [122].

The protein-based systems for nanodelivery of TRAIL pathway-targeting ligands, which are described above, are summarized in Table 4.

## 6. Scaffolds for Multimerization of DR5-Targeted TRAIL-Mimicking Peptide

In addition to the TRAIL ligand and TRAIL death receptor-targeting agonistic antibodies, a set of recently developed nanodelivery systems are designed to carry the DR5-targeted peptide, an apoptogenic cyclic pentadecapeptide mimetic of TRAIL, which specifically recognizes DR5 [123]. It is worth including in a pool of TRAIL pathway-targeting nanovehicles because of their common purpose of action; however, it stands apart due to its truncated structure and, therefore, insufficient functionality. Despite competing with TRAIL upon binding to the DR5 receptor, the DR5-targeted peptide did not induce apoptosis as a solitary ligand [124]. However, DR5-targeted peptide has been extensively exploited for conjugation with a set of specific multimerization scaffolds, the first of which were the adamantine-based dendrons.

Dendrimers and dendrons can serve as a scaffold for the multimerization of ligands due to their nanoscale size, monodispersity, and the ability for modification by functional groups. The adamantane dendritic structures provide a rigid tetrahedral arrangement of the attached moieties. Such a tripodal structure is perspective for the recognition by r m a receptors, which perform their function in the trimeric form, such as TRAIL death receptors. When the adamantine-based dendrons were explored as multivalent scaffolds for the DR5-targeted peptide, it was demonstrated that the multivalent form of the peptide is necessary to trigger a substantial DR5-dependent apoptotic response; only the trimeric and hexameric ligands were able to induce cell death in DR5-expressing BJAB cells. Unexpectedly, the hexamerized peptide ligand displayed a decreased biological activity relative to the trimerized one, probably due to the steric hindrance leading to unfavorable interactions between the peptides and DR5 receptor on the cell surface. The authors speculated that the insertion of a longer spacer between the adamantane units might render the functional groups more accessible, thus generating higher multimeric structures [125].

Another research group similarly introduced a modular approach to generate DR5-binding constructs, comprising multiple copies of the same DR5-targeting peptide [123] (named here DR5TP), covalently bound to biomolecular scaffolds of a peptidic nature. The authors designed the macromolecular architectures based on a scaffold of a C-terminal oligomerization domain of human C4b binding protein (C4BP), which is a plasma component without biological function, allowing the assembly of seven DR5TP units. To localize the tethered DR5-binding motifs in the target scaffold, the enzyme recognition sites were introduced, and DR5TP-based fragments were attached by an enzyme-catalyzed coupling. The resulting heptameric constructs showed remarkable pro-apoptotic activity when DR5TP was ligated to the C-terminus. Supporting the previous data, this indicates that the inter-ligand distance, relative spatial orientation, and copy number of receptor-binding moieties are prerequisites for the DR5 binding capacity and apoptotic cell death [126].

This research was continued further by the covalent multimerization of the DR5TP on a polysaccharide dextran scaffold. Dextran is a biocompatible polysaccharide synthesized by lactic acid bacteria and consisting of repeating a-(1–6)-linked oligo-d-glucose units, which is accredited as a blood-flow enhancer and plasma expander and is generally regarded as safe by the FDA. The attachment of multiple DR5TP ligands on a flexible scaffold exhibited efficient killing of tumor cells Colo205 and Jurkat mediated by DR5 clustering. Importantly, the dextran framework introduced high conformational flexibility, which compensated for the need for the defined ligand orientation that was earlier considered essential for effective receptor clustering and apoptosis induction [127].

The multimerization of the DR5-targeted peptide has been further advanced by the development of supramolecular self-assembling peptide amphiphile nanostructures, in some sense approximating them to polymeric micelles. Here, a TRAIL-mimetic peptide sequence was incorporated into a peptide amphiphile (PA), which forms one-dimensional nanofibers by supramolecular self-assembly. These PA-nanostructures are non-specifically cytotoxic to cancer cells even without specific targeting and are also capable of delivering small molecule drugs [128]. The previously reported TRAIL-mimetic peptide sequence [124] was covalently attached to the PA, creating a PA with two DR5 binding sequences, which formed cylindrical nanofibers upon mixing with PEGylated PA. The nanostructures that externally display TRAIL-mimetic peptides with encapsulated paclitaxel exhibited enhanced in vivo efficacy in a breast cancer model, demonstrating that the two therapies are effective when combined within a single nanostructure specifically targeted to cancer cells [129]. Therefore, it can serve as an example of multifunctional peptide-based supramolecular nanotherapeutics.

However, the above-mentioned strategies do not allow for precise control of the spatial presentation of the targeting ligands. Therefore, another group of authors attempted to conjugate the same DR5-targeting peptide with DNA origami nanostructures, which provide strict orientation and inter-ligand distance of the patterns of peptides displayed in hexagons. Importantly, it was revealed that effective triggering of DR5 clustering needs around 5 nm-spaced hexagonal ligand patterns. The obtained DNA origami presenting TRAIL mimicking peptide patterns with sub-10 nm inter-ligand spacing induced effective DR5 clustering and strong apoptosis induction in both TRAIL-sensitive and TRAIL-resistant breast cancer cells. It is speculated that the ability of the monomeric peptide binders to trigger apoptosis could indicate that the pre-ligand clustering state of DR5 is less important than previously thought and that the clustering of DR5, whatever the pre-ligand state may be, is enough to trigger apoptosis [130]. The above-mentioned studies provide a possible explanation for the frequently low efficacy of the ligand- and antibody-based methods of DR5 activation. Importantly, the studies of the multimerization of DR5-targeted TRAIL-mimicking peptides on various scaffolds provide valuable data for the mechanism of TRAIL death receptor clustering by the nanodelivery systems carrying specific ligands described throughout this review. This can contribute to the further development of effective nanotherapeutics targeting the TRAIL pathway for the treatment of various diseases.

The scaffolds for the multimerization of the DR5-targeted TRAIL-mimicking peptide described above are summarized in Table 5.

## 7. Discussion

Since the TRAIL-dependent pathway is universal for most types of transformed cells, there appears to be no preference for any particular type of tumor. Therefore, the developed nanosystems targeting the TRAIL pathway are aimed at various types of tumors, including fibrosarcoma, melanoma, acute myeloid leukemia, lung, colorectal, breast, ovarian, and pancreatic ductal adenocarcinomas, etc. Additionally, a few works on other diseases, such as rheumatoid arthritis, cystic fibrosis, and Alzheimer’s disease, attest to the versatility of this mechanism and the potentially wide range of activity of nanodelivery systems modified with TRAIL pathway-targeting ligands in the body.

The nanodelivery of the ligands targeting the TRAIL pathway allows the achievement of several goals. First, conjugation with nanocarriers helps to improve the initially poor pharmacokinetic parameters of the ligand, in particular TRAIL. For example, it was demonstrated that TNM-PN, TRAIL-modified paclitaxel-loaded NM camouflaging nanoparticles, obtained a prolonged blood circulation time [85], gel-like mPEGylated coacervate (mPEG-Coa) delivery platform bearing cargo TRAIL provided a long-term sustained release [88], and TRAIL-presenting ferritin nanocages TTPNs obtained improved pharmacokinetic characteristics and stability [106]. However, more importantly, the multimerization of the ligand on the surface of nanoparticles mimics the naturally occurring membrane-bound TRAIL and contributes to the effective receptor clustering and enhanced apoptosis induction [38,131]. Additionally, death-receptor-targeting ligands usually acted synergistically with various antitumor compounds, such as chemotherapeutics and photosensitizers. Therefore, the complex nanodelivery systems, which contain a combination of the aforementioned drugs, exhibit enhanced antitumor activity.

Another important issue is that nanocarriers encapsulating low molecular weight therapeutics, including those commercially available, are usually aimed at improving pharmacokinetics and reducing side effects, which are often caused by chemotherapy drugs. The functionalization with ligands targeting the TRAIL pathway allows for tumor-specific targeting of such combined nanotherapeutic agents, which, together with the aforementioned synergistic effect, results in increased efficacy.

Among the nanodelivery systems reviewed, a few are only proof-of-concept studies with preliminary data on potential efficacy in vitro. The therapeutic efficacy of most nanosystems targeting the TRAIL pathway has been demonstrated in vivo, mainly in murine tumor xenograft models (for those with anticancer applications). However, despite the huge variety of nanodelivery systems modified with TRAIL pathway-targeting ligands developed in recent years, none of them have yet reached clinical trials. One of the obstacles to clinical implementation is the complexity of achieving uniformity in large-scale manufacturing and acceptable batch-to-batch variation of combined multicomponent nanosystems containing several substances with their own unique properties. Another is the need to consider not only the mechanism of action but also the possible side effects of all components as well as take into account potential new properties that may appear when they are combined. From this follows the complexity of the regulatory framework for admission to clinical trials and approval of such complex nanopreparations, serving as one of the reasons for the relatively small number of such drugs on the market.

Since the reviewed nanodelivery systems functionalized with TRAIL pathway-targeting ligands are composed of materials that have very different properties, they are poorly eligible for direct comparison. However, their prospects can be assessed on examples of similar nanosystems that are already on the market. The well-studied materials, which are approved by the FDA or EMA for therapeutic applications, have a good chance of clinical implementation due to the large amount of accumulated data on their properties. The majority of the nanocarriers, which are already approved for therapeutic application, particularly for cancer treatment, are liposomes intended for sustained release or reduced systemic toxicity of low molecular weight chemotherapy drugs, such as Onivyde^®^ for irinotecan, Lipusu^®^ for paclitaxel, or Vyxeos^®^ for daunorubicin and cytarabine [2]. However, among the marketed nanodelivery systems with a recombinant protein as an active component, polymer-based nanosystems are prevailing, for example, Oncaspar^®^ for L-asparaginase, Neulasta^®^ for filgrastim, Plegridy^®^ for recombinant IFN-β, Cimzia^®^ for TNF-α-specific IgG Fab’ fragment, and Adynovate^®^ for coagulation factor VIII [2]. Accordingly, polymer-based nanocarriers constitute the largest group among those functionalized with TRAIL pathway-targeting ligands (Table 3). Therefore, we can speculate that this type of nanodelivery system has better prospects for clinical implementation. The inorganic nanoparticles, despite their unique properties and capacity of dual application for theranostics, make up a smaller group of marketed nanocarriers. This is primarily due to poorer biocompatibility, which forces them to further address systemic toxicity issues. However, novel developments, such as protein nanocages, virus-based scaffolds, nanogels, and DNA origami, obviously have a huge innovative potential and may outperform well-known materials in the future, provided they are thoroughly investigated.

## 8. Conclusions

Currently, there are more than 50 clinically approved nanomedicines, most of which are based on simple designs consisting of a small number of components. The synthesis of more complex nanoparticles that resemble biological structures with hundreds of functional components is often poorly compatible with large-scale clinical-grade manufacturing. On the one hand, simple systems have an advantage for large-scale production over complex ones due to easy handling, reproducibility, and cost-effectiveness. On the other hand, the innovative «smart» composite nanovehicles are capable of combining a set of controversial properties aiming to reach the desired therapeutic effects. Therefore, when designing innovative nanotherapeutics, researchers need to maintain a reasonable balance between efficiency, productivity, and sustainability.

The design of novel nanovehicles which target the TRAIL pathway faces a number of challenges. The first concerns such issues as safety, biocompatibility, specific targeting, low immunogenicity, cost- and time-effective large-scale clinical-grade manufacturing, and sustained release obtained with different types of nanodelivery systems. The second addresses the geometric and steric aspects of ligand multimerization for efficient death receptor clustering and apoptosis induction. Finally, the most important outcome is the obtained therapeutic effect.

Unfortunately, the therapeutic effect achieved by the different nanodelivery systems is difficult to compare directly due to their different natures and compositions, as well as the different stages of research to which they have progressed. We can generally conclude that the encapsulation of ligands targeting the death receptor pathway into nanovehicles usually results in enhanced efficacy in various disease models, tumors in particular. The initially low efficacy due to insufficient clustering of the target receptors can be overcome by multimerization of the ligands on the surface of nanocarriers of different natures using a variety of physical and chemical methods, whereby ligand immobilization on the surface of nanoparticles in a strict orientation and a fixed distance between the molecules seems to be a significant advantage. This may be useful for the future design of nanotherapeutics whose effects are based on the effective clustering of target receptors. Additionally, the synergistic antitumor effect of nanoconjugated ligands targeting the TRAIL pathway with various physical and chemical effectors plays an important role in boosting performance. Therefore, a large number of developments based on both well-studied and innovative nanomaterials give hope that nanoscale delivery systems carrying TRAIL pathway-targeting ligands will be implemented in clinical practice.

## Figures and Tables

**Figure 1 pharmaceutics-15-00515-f001:**
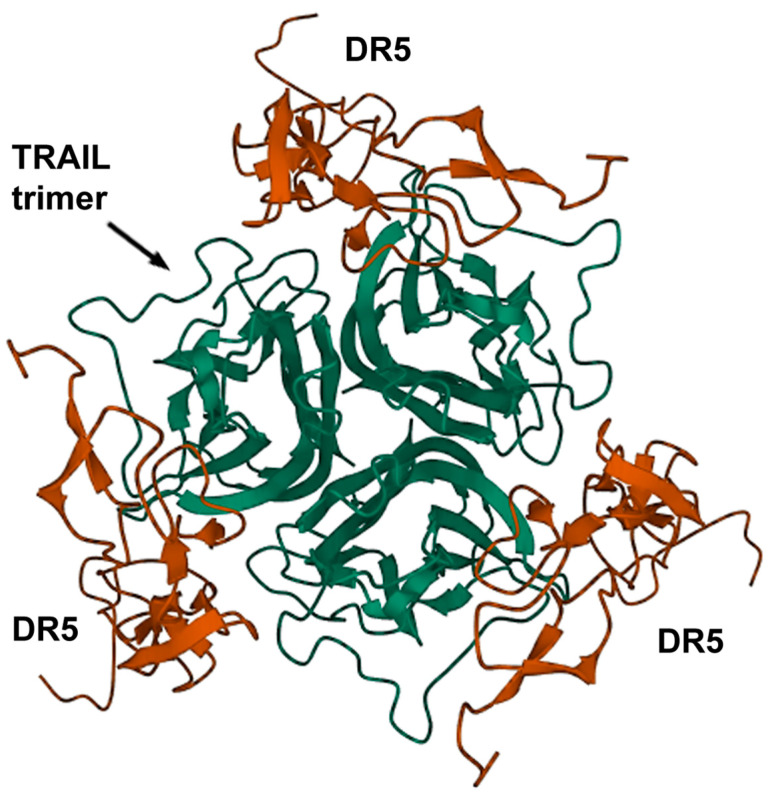
Three-dimensional structure of TRAIL in complex with the DR5 receptor retrieved from Protein Data Bank (PDB number 1D4V).

**Figure 2 pharmaceutics-15-00515-f002:**
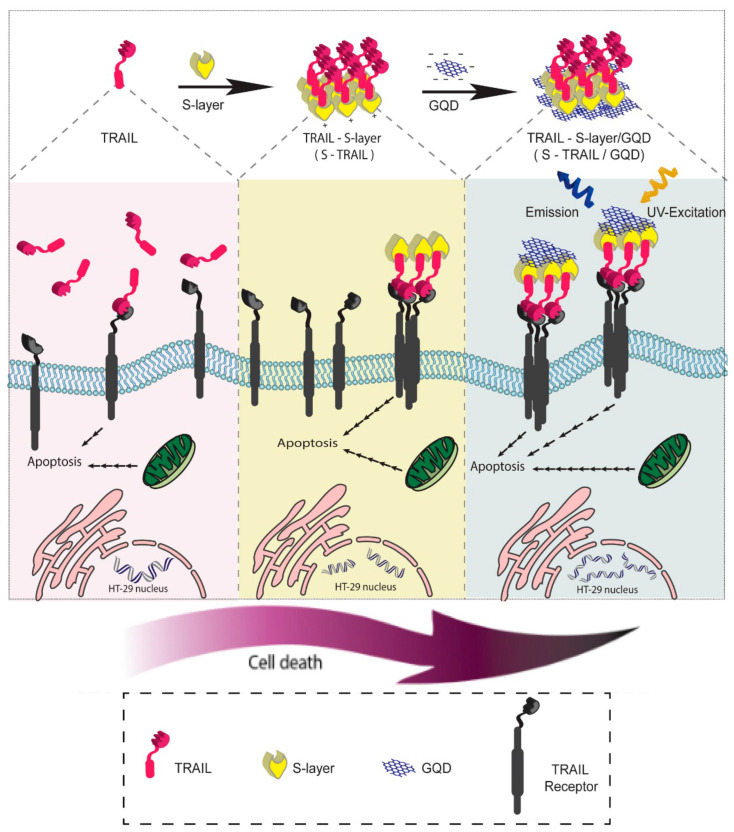
Schematic illustration of S-TRAIL and S-TRAIL/GQD preparation and biological function. Reproduced from [34]. Copyright © 2022, The Authors (CC BY 4.0 http://creativecommons.org/licenses/by/4.0/ (accessed on 20 January 2023)).

**Figure 3 pharmaceutics-15-00515-f003:**
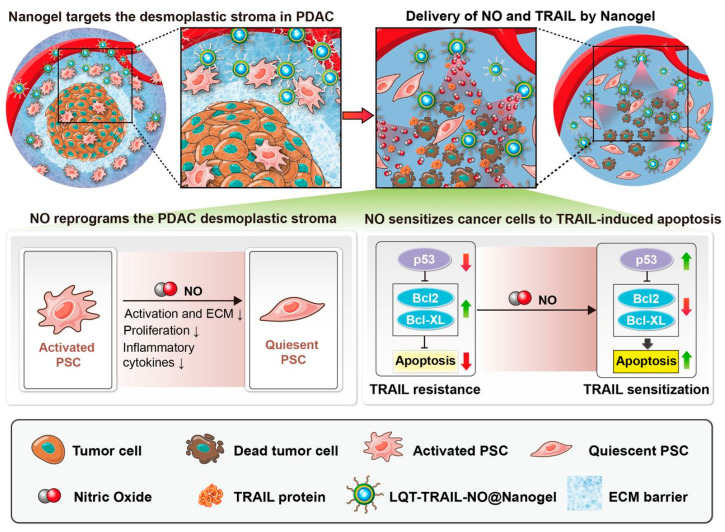
Schematic showing the mechanism by which tumor stroma-targeted TRAIL-NO@Nanogel suppresses PDAC progression in mice. NO released from tumor stroma-targeted TRAIL-NO@Nanogel remodels the fibrotic tumor microenvironment of desmoplastic PDAC. (1) NO released from NPs modified with tumor stroma-targeting peptides identified by phage display suppresses PSC activation, reduces ECM production, and increases tumor perfusion in PDAC. (2) NO reprogrammed the desmoplastic stroma and overcomes TRAIL resistance, sensitizing PDAC tumors to TRAIL therapy. (3) Co-delivery of TRAIL and NO by tumor stroma-targeted TRAIL-NO@Nanogel efficiently suppresses tumor growth. ECM, extracellular matrix; NO, nitric oxide; NPs, nanoparticle; PDAC, pancreatic ductal adenocarcinoma; PSC, pancreatic stellate cells; TRAIL, tumor necrosis factor-related apoptosis-inducing ligand. Reproduced with permission from [89]. Copyright © 2022, BMJ Publishing Group Ltd. & British Society of Gastroenterology (CC BY-NC 4.0 http://creativecommons.org/licenses/by-nc/4.0/ (accessed on 20 January 2023)).

**Table 1 pharmaceutics-15-00515-t001:** Structure, effect, and research stage of the inorganic nanoparticles modified with TRAIL pathway-targeting ligands.

Material Type	Name	Structure	Effect	Stage	Ref.
**Metal**					
Gold	Co-Ox-AuNPs	Gold NPs with an outer lipid layer comprising a free carboxyl group encapsulated with oxaliplatin and surface modified by the DR5-targeted antibody, containing an amine group, via the carbodiimide chemistry.	Synergistically reduced xenografts HCT-116 tumor growth.	In vivo	[15]
	Nanogold-TRAIL complexes	Citrate-coated gold NPs conjugated with the recombinant human TRAIL.	Anti-inflammatory and pro-tumorigenic activities in M2 and tumor-associated macrophages. Increased TRAIL cytotoxicity in M2-polarized macrophages.	In vitro	[16]
Silver	TRAIL-AgNPs	Citrate-reduced silver NPs with non-covalently absorbed TRAIL protein.	Increased cytotoxicity in TRAIL- resistant derivatives of human glioblastoma T98G cells.	In vitro	[18]
	AgCTP NPs	Silver NPs conjugated with TRAIL and coated with PEG.	Inhibited proliferation and colony formation of colon cancer HT-29 cells.	In vitro	[19]
Iron oxide	NC@TRAIL	TRAIL-grafted iron oxide nanoclusters (NCs) functionalized with (3-aminopropyl)triethoxysilane (APTES).	Induced breast cancer MDA-MB-231 cell death during the thermal treatment by generating MHT and PT-initiated hotspots at the cell surface.	In vitro	[24]
	NV10 NV100	TRAIL-grafted magnetic spinel iron oxide NPs of defined core size 10 and 100 nm.	Enhanced apoptosis rate in HCT116 and HepG2 cells.	In vitro	[25]
	NH-TRAIL@NPs-CO CO-TRAIL@NPs-NH	Maghemite NPs functionalized with amino groups and conjugated with carboxylic acid groups of TRAIL, or NPs functionalized with carboxylic acid groups and conjugated with the amino groups of TRAIL.	Greater pro-apoptotic effect of the NH-TRAIL@NPs-CO than the CO-TRAIL@NPs-NH in human breast and lung carcinoma cells.	In vitro	[26]
	NanoTRAIL	Iron oxide NPs with electrostatically immobilized TRAIL on the surface.	Improved survival outcomes in a colorectal cancer patient-derived xenograft model compared with TRAIL monotherapy.	In vivo	[27]
**Carbon**					
Graphene	TRAIL/DOX-fGO	DOX-bearing graphene oxide (GO) nanosheet, connected by PEG linker with the furin-cleavable peptide, conjugated with TRAIL by sulfhydryl groups.	Synergistic antitumor activity in the xenograft A549 tumor-bearing nude mice.	In vivo	[29]
	GO+A+S	Graphene oxide (GO) covalently linked with the DR4 antibody, decorated with 1-pyrenemethylamine hydrochloride (PyNH_2_) and AKT siRNA.	Synergistic inhibition of the growth of xenograft MCF-7 tumors in a mouse model and enhanced therapeutic effect.	In vivo	[32]
	S-TRAIL/GQD	Graphene quantum dots (GQDs) conjugated non-covalently with TRAIL fused with crystalline bacterial cell surface layer protein.	The anticancer efficacy in TRAIL-resistant HT-29 human colon carcinoma cells.	In vitro	[34]
Carbon nanotubes	NPT	Single-walled carbon nanotubes (SWCNTs) vectorizing TRAIL via non-covalent 1-pyrenebutanoic acid N-hydrosuccinimide ester (PSE).	Nearly 20-fold increase of TRAIL pro-apoptotic potential in human tumor cell lines of different origin	In vitro	[36,37,39]
**Boron nitride**					
Boron nitride nanotubes	NPT	Boron nitride nanotubes (BNNTs) anchoring TRAIL via non-covalent 1-pyrenebutyric acid N-hydroxysuccinimide ester (PSE) and mixed with distearoyl-sn-glycero-3-phosphoethanolamine-N- conjugates.	Enhanced binding affinity to TRAIL receptors DR4 and DR5		[41]
**Silica**					
Periodic mesoporous organosilica	DOX@PMO-hT	DOX-loaded periodic mesoporous organosilica (PMO) nanoparticles surface-modified by maleimide and conjugated with cysteine-modified TRAIL fused with the C-terminus fiber shaft of human adenovirus type 5 (HA5ST).	Synergistic induction of TRAIL-dependent cell death and immunogenic cell death in tumors together with activation of immune cells through the immunomodulatory functions of PMOs.		[43]

**Table 2 pharmaceutics-15-00515-t002:** Structure, effect, and research stage of the lipid-based systems for nanodelivery of TRAIL pathway-targeting ligands.

Name	Structure	Effect	Stage	Ref.
**Liposomes**				
LUVDOX-TRAIL	DOX-loaded large unilamellar vesicles (LUV) with surface-anchored soluble TRAIL.	Enhanced activity in various tumor models, including the concerted action in a fibrosarcoma HT1080-xenograft model without systemic toxicity.	In vivo	[45,46,47,48,49,50]
ES/TRAIL	Liposomes surface-conjugated with TRAIL and the adhesion receptor E-selectin (ES).	Neutralization of circulating tumor cells (CTCs) in the bloodstream; prevention of the spontaneous formation of metastatic tumors in an orthotopic xenograft model of prostate cancer; reduction of metastasis in triple-negative breast cancer (TNBC) mouse model; synergistic apoptotic effect in prostate cancer cells with a liposomal form of piperlongumine.	In vivo	[51,52,53,54,55]
NK cell-liposome conjugate	TRAIL- and anti-CD57 monoclonal antibody-functionalized liposomes conjugated with CD57-expressing NK cells.	Significant apoptosis induction in cancer cells of various origins.	In vitro	[56]
“super” NK cells	TRAIL- and anti-NK1.1 antibody- functionalized liposomes conjugated with NK cells.	Enhanced retention time within the tumor-draining lymph nodes; prevention of the lymphatic spread of a subcutaneous tumor in mice	In vivo	[57]
TRAIL-[Lip-PTX]^C18-TR^	Paclitaxel (PTX)-loaded liposomes with electrostatically attached TRAIL and a stearyl chain (C18) fused with histidine-rich cell-penetrating peptide (CPP) TR.	Improved drug release profile, stability, circulation time in vivo, and tumor suppression effect in a melanoma model.	In vivo	[59]
pPB-SSL-TRAIL	Sterically stabilized liposomes targeted for PDGFR-β by the specific affinity cyclic peptide C*SRNLIDC* (pPB), bearing TRAIL as a cargo.	Increased apoptosis of aHSCs and remarkably alleviated hepatic fibrosis in mice.	In vivo	[61]
**Solid lipid nanoparticles (SLN)**				
DR5-DAPT-SLNs	SLN surface-modified with DR5 antibody and entrapping γ-secretase inhibitor (GSI) DAPT.	Enhanced cellular uptake and cytotoxicity compared to non-targeted SLNs; greater tumor regression in vivo compared to DAPT-SLNs and DAPT alone.	In vivo	[62]
C I-p/Q NLC	Nanostructured lipid carriers (NLC) decorated with DR5-targeted antibody conatumumab (C) and co-encapsulated with irinotecan prodrug (I-p) and a natural bioflavonoid quercetin NLC (Q).	Higher cytotoxicity than non-decorated NLC, single drug-loaded NLC, and free drugs on colorectalcancer (CRC) cells in vitro and in xenografts mouse models of CRC in vivo.	In vivo	[63]

**Table 3 pharmaceutics-15-00515-t003:** Structure, effect, and research stage of the polymer-based systems for nanodelivery of TRAIL pathway-targeting ligands.

Name	Structure	Effect	Stage	Ref.
**Natural polysaccharides**				
TRAIL/ALG-CaCO_3_	Porous DOX-loaded CaCO_3_ core electrostatically functionalized with TRAIL and an alginate (ALG) multilayer shell by a layer-by-layer (LbL)-technique.	Notable anticancer activity in HeLa cells vitro.	In vitro	[66]
GCS-aDR5 scFv	Glycol chitosan nanoparticles (GCS) loaded with anti-human DR5 single-chain antibody (aDR5 scFv) by ion gelation.	Efficient targeting to the tumors rather than healthy organs; inhibition of hepatocellular H22 xenograft tumor growth.	In vivo	[67]
OIHNPs	Chitosan polymer core with encapsulated oxaliplatin, coated by the outer negative charged lipid layer and covalently conjugated with thiolated anti-TRAIL(CD-253) antibody.	Sustained release and effective reduction of tumor volume in xenograft HT-29 tumor models in vivo.	In vivo	[68]
DR5 TMP NP	Chitosan/alginate nanoparticles with encapsulated photosensitizer meso-Tetra(N-methyl-4-pyridyl) porphine tetratosylate (TMP) conjugated with anti-DR5 antibody AMG655 via carbodiimide chemistry.	Enhanced effectiveness of entrapped TMP in colorectal carcinoma HCT116; enhanced activation of caspase 8 compared with free AMG655.	In vitro	[69]
**PLA, PLGA**				
DR5-NP CPT DR5 NP	PLGA nanoparticles conjugated with anti-DR5 antibody AMG 655 via carbodiimide chemistry and loaded with DNA topoisomerase 1 inhibitor camptothecin (CPT).	Efficient DR5 receptor clustering leading to activation of apoptosis in colorectal models; overcoming resistance to TRAIL-induced apoptosis in HCT116 colorectal cancer model.	In vivo	[72,73]
αDΡ5-NP CPT DR5 NP	NHS-modified PLGA-PEG was incorporated in the polymer blend, conjugated in one step to the AMG 655 (αDΡ5-NP), and loaded with camptothecin (CPT DR5 NP).	Improved scalability of the fabrication; enhanced ability to induce apoptosis in pancreatic cancer models.	In vivo	[74]
PEG-PLA DOX TRAIL MB	Polymeric microbubbles (MBs) of polylactic acid (PLA) with inserted polyethylene glycol (PEG), covalently functionalized by TRAIL via maleimide chemistry and co-encapsulated with DOX.	Increased efficiency in human breast adenocarcinoma cells by directing the drug-loaded nanoshards into the tumor in response to ultrasound focused at the site.	In vitro	[76,77,78]
PLGA/sTRAIL nanofibers	Encapsulating TRAIL into PLGA nanofibers by coaxial electrospinning.	Inhibition of xenograft breast tumor growth.	In vivo	[81]
NANO-A NANO-B	TRAIL-neutralizing monoclonal antibody adsorbed on the surface of polymeric (PLGA, NANO-A) and lipidic (NLC, NANO-B) nanoparticles.	Reaching the brain through the nose-to-brain route and localizing in hippocampal areas of Alzheimer’s diseased mice models.	In vivo	[83]
TU-NPs	PLGA cores loaded with hydroxychloroquine (HCQ) and coated by the plasma membrane of umbilical vein endothelial cells stably expressing TRAIL.	Targeting the activated M1 macrophages in the inflammatory sites of the collagen-induced arthritis (CIA) mouse model.	In vivo	[84]
TNM-PN	PLGA core preloaded with paclitaxel (PTX), camouflaged with a neutrophil membrane (NM), and modified with TRAIL.	Prolonged blood circulation; preferential tumor accumulation; SKOV3 tumor inhibition; enhanced animal survival.	In vivo	[85]
CPT MV	PLGA core loaded with a photosensitizer chlorin e6 (CP NPs) and coated with a TRAIL-expressing cell membrane vehicle (TRAIL MV) from lentiviral murine TRAIL-expressing CHO-S cells.	Production of ROS in the targeted cells upon laser irradiation; suppression of xenograft murine hepatoma hepa 1-6 tumors; high biosafety.	In vivo	[86]
**Nanogels**				
TRAIL-loaded mPEG-Coa	TRAIL incorporated into gel-like colloidal droplet coacervate, obtained by the electrostatic interaction of the mPEG-conjugated poly(ethylene arginyl aspartate diglyceride) (PEAD) with heparin, followed by self-assembly in aqueous conditions.	Long-term sustained release; suppression of HCT116 colon cancer cells recurrence.	In vitro	[88]
TRAIL-NO@Nanogel	TRAIL-loaded silk fibroin (SF) hydrogel core covered by the shell composed of lipids and PLGA and loaded with a synthetic NO donor dinitrosyl iron complex (DNIC; Fe(μ-SEt)_2_(NO)_4_).	Enhanced antitumor efficacy in spheroid cultures in vitro and in orthotopic PDAC models in vivo.	In vivo	[89]
TRAIL-nanogel	TRAIL encapsulated in a bactericidal crosslinked cationic poly(L-lysine)-block-poly(L-threonine) (PLL-b-PLT) co-polypeptide nanogel.	Protection of mice against *K. pneumoniae*- induced sepsis; prolonged survival in septic mice; reduced bacterial numbers in circulation.	In vivo	[90]
**Micellar nanoparticles**				
TRAIL-M-CTX	Polymeric self-assembled micelles from amphiphilic copolymers monomethoxyl poly(ethylene glycol)–b-poly(DL-lactide) (mPEG-PLA) and COOH-PEG-PLA, loaded with cabazitaxel (CTX) and surface modified with TRAIL via carbodiimide chemistry.	Improved anticancer efficacy with synergistic effects against TRAIL-resistant human breast cancer MCF-7 cells.	In vitro	[93]
TRAIL-Cur-NPs	Polymeric nanoparticles composed of PEG as a hydrophilic segment and poly(e-caprolactone) (PCL) as a hydrophobic segment, loaded with curcumin and coated with TRAIL via electrostatic interactions.	Prolonged circulation time; enhanced cellular uptake; superior therapeutic effect on HCT116 colon cancer cells in a xenograft mouse model in vivo.	In vivo	[94]
IPN@TRAIL	Iron oxide nanoparticles encapsulated into the core of micelles, composed of self-assembled PEGylated cationic amphiphilic polymers (Alkyl-PEI2k-PEG2k) photothermal agent metalla-aromatics complex Ph556, and negatively charged TRAIL adsorbed via electrostatic interactions.	Improved synergistic killing effect between PTT and TRAIL in a controllable manner on TRAIL-resistant A549 lung tumor model-bearing mice.	In vivo	[95]
PVP-DR5-B	Amphiphilic polymeric poly(N-vinylpyrrolidone) (Amph-PVP) micelles, self-aggregating in aqueous solutions and covalently conjugated with DR5-specific TRAIL variant DR5-B.	Enhanced cytotoxicity on multicellular tumor spheroids of HCT116, HT29, and MCF-7 cells without cytotoxicity to normal cells.	In vitro	[98]

**Table 4 pharmaceutics-15-00515-t004:** Structure, effect, and research stage of the protein-based systems for nanodelivery of TRAIL pathway-targeting ligands.

Name	Structure	Effect	Stage	Ref.
**Protein nanoparticles**				
GTB NPs	BSA core loaded with gambogic acid (GA), adsorbed by polyethylenimine, and shielded with carboxymethyl chitosan- folate (CMCS-FA) as the outer shell for encapsulating TRAIL protein by the layer-by-layer (LbL) technique.	Improved antitumor efficacy in the MCF-7 xenograft model; reduced side effects.	In vivo	[100]
TRAIL/ALG-DOX@BSA	DOX-loaded BSA nanoparticles encapsulating oppositely charged TRAIL and alginate by consecutive adsorption using the layer-by-layer technique.	Synergistic cytotoxic effect in H460 cells and in the DOX-resistant L929 cells.	In vitro	[101]
TRAIL-FF	Nanoparticles composed of TRAIL and a diphenylalanine (FF) dipeptide via π–π stackings and hydrogen bonds through supramolecular self-assembly.	Cytotoxic effect on the MCF-7 and H460 cancer cells.	In vitro	[102]
RGD-TRAIL-ELP	Recombinant RGD-TRAIL fused to the temperature-sensitive elastin-like polypeptide (ELP), self-assembling into nanoparticles at 37 °C under physiological conditions.	Enhanced tumor regression in the COLO-205 tumor xenograft model.	In vivo	[103]
**Protein cages**				
TTPNs	24 human ferritin heavy chain (hFTH) monomeric subunits self-assembling into the cage structure, genetically fused with TRAIL displayed on the surface of nanocages.	Enhanced stability; improved pharmacokinetic characteristics and antitumor activity in a HepG2 xenograft mouse model.	In vivo	[106]
DOX-TTPNs	TTPNs with Cu-DOX complex loaded into the inner cavity of ferritin nanocage through its metal ion storage mechanisms.	Re-sensitizing TRAIL-resistant xenograft tumors of colorectal cancer cell line HT29 in vivo.	In vivo	[107]
TRAIL-ATNC TRAIL-ATNC^IL4rP^	Ferritin nanocage presenting TRAIL in trimer-like conformation by connecting to the triple helical domain of pulmonary surfactant-associated protein D (TRAIL-ATNC), ligated to IL4R binding peptide (TRAIL-ATNC^IL4rP^).	Enhanced antitumor efficacy in an orthotopic pancreatic cancer model.	In vivo	[108]
TRAIL-P(RGD) NC	TRAIL covalently conjugated to the surface of the self-assembled proteinoid nanocapsules (NC) based on D-arginine®, glycine (G), and L-aspartic acid (D).	Cytotoxicity in human ovarian cancer cells.	In vitro	[110]
AaLS/TRAIL/EGFRAfb	Genetically engineered lumazine synthase protein cage nanoparticle (AaLS) presenting 60 STs (split proteins derived from the second immunoglobulin-like collagen adhesin domain from Streptococcus pyogenes) on its surface, with human TRAIL and EGFR binding affibody molecules (EGFRAfb) genetically fused to SC and polyvalently displayed on the surface using an ST/SC post-translational protein-ligation system.	Improved cytotoxic effect against EGFR-overexpressing cancer in vitro and in vivo.	In vivo	[113]
**Virus-based scaffolds**				
PVXHisTRAIL	Elongated PVX nanoparticle displaying up to 490 TRAIL protein molecules, non-covalently conjugated by an N-terminal His-tag with nickel-coordinated nitrilotriacetic acid (NiNTA) modules displayed on PVX surface.	Delayed tumor growth in an athymic nude mouse model bearing human triple-negative breast cancer xenografts in vivo.	In vivo	[117]
A4 ZA4	TRAIL-linked oncolytic adenovirus rAd5pz-zTRAIL-RFP-SΔ24E1a (A4), coupled to capsid protein IX (pIX) by a synthetic leucine zipper-like dimerization domain (zipper) and coated with additional zipper-fused TRAIL (ZA4).	Inhibited proliferation of acute myeloid leukemia cells from cell lines and primary patient samples in vitro; enhanced antitumor activity in vivo.	In vivo	[120,121]
P-VMNs	Virus-mimetic nanocapsules (VMNs) composed of TRAIL, electrostatically entrapped onto the artificial capsids (AC), coated with envelopes of positively charged chitosan oligosaccharides (COS), and decorated with Arg-Gly-Asp (RGD) peptides.	Deep tumor penetration and accumulation; antitumor efficacy in chemoresistant human colorectal carcinoma LoVo/R tumor-bearing mice.	In vivo	[122]

**Table 5 pharmaceutics-15-00515-t005:** Structure, effect, and research stage of the scaffolds for multimerization of DR5-targeted TRAIL-mimicking peptide.

Name	Structure	Effect	Stage	Ref.
Trimer 4 Hexamer 10	DR5-targeted TRAIL-mimicking peptide chemically conjugated with adamantine-based dendrons to form trimeric and hexameric structures.	Induction of cell death in DR5-expressing BJAB tumor cells.	In vitro	[125]
C4BP-DR5TP	DR5-targeting peptide (DR5TP) covalently ligated to C-termini of heptameric constructs based on a short (60–75 residues) scaffold of a C-terminal oligomerization domain of human C4b binding protein (C4BP).	Induction of cell death in Colo205 tumor cells.	In vitro	[126]
DR5TP-loaded dextran	Covalent multimerization of the DR5TP on a polysaccharide dextran scaffold.	Induction of cell death in Colo205 and Jurkat tumor cells.	In vitro	[127]
DR5-binding PA	TRAIL-mimetic peptide covalently attached to a peptide amphiphile (PA), forming one-dimensional nanofibers by supramolecular self-assembly.	Enhanced efficacy in a breast cancer model in vivo.	In vivo	[129]
Peptide-decorated DNA origami	DR5-targeting peptide conjugated with DNA origami nanostructures by click chemistry.	Strong apoptosis-induction in human breast cancer cell lines MDA-MB-231, MCF-7, and SK-BR-3.	In vitro	[130]

## Data Availability

No new data were created or analyzed in this study. Data sharing is not applicable to this article.

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
