# Peer review of "Recent Advances in the Development of Nanodelivery Systems Targeting the TRAIL Death Receptor Pathway"

_pharmaceutics, 2023, doi:10.3390/pharmaceutics15020515_

Round 1

Reviewer 1 Report

In this review, the author introduced recent research advances in nano-delivery systems targeting the TRAIL death receptor pathway. This work is instructive for drug development in this area. However, there are some writing and presentation issues in the manuscript that I would like to suggest to the authors for refinement.

1.      In this review, the authors do not describe the methodology of the study, nor do they mention the way in which the literature was collected and the scope of its inclusion. I would suggest that the authors add this information in the introduction section.

2.      The author list various drug delivery systems in the text, but did not compare and analyze them, what their advantages and disadvantages were, and which ones have better prospects for clinical application. I would suggest that this be added to the discussion section.

3.      These drug delivery systems were mentioned in this review. Whether they are better selective for certain tumors, or on which tumors the research on these delivery systems has been mainly conducted. It is suggested that the authors summarize the relevant information in the discussion section.

4.      I would suggest that the authors clearly list the current stage of research for these delivery systems, for example whether they are in vitro studies, or in vivo studies, or clinical trials.

5.      To help the reader understand, the first occurrence of the abbreviation should be marked with the full name. For example, "ES/TRAIL" in line 300.

6.      Please confirm whether the content of line 341-343 and 381-384 require references.

7.      Please check and correct the spell that needs to be superscripted throughout the text. Such as Line 172, “Fe2+” should be “Fe2+”.

Author Response

Thank you for your valuable comments. They helped to improve the manuscript significantly.

  1. In this review, the authors do not describe the methodology of the study, nor do they mention the way in which the literature was collected and the scope of its inclusion. I would suggest that the authors add this information in the introduction section.

Response. The literature for the narrative review was collected from publicly available databases (Web of Science, Scopus, PubMed, etc.). We have added the description of methodology and the criteria for the inclusion to the Introduction section.

  1. The author list various drug delivery systems in the text, but did not compare and analyze them, what their advantages and disadvantages were, and which ones have better prospects for clinical application. I would suggest that this be added to the discussion section.

Response. We have added the Discussion section with the appropriate information.

  1. These drug delivery systems were mentioned in this review. Whether they are better selective for certain tumors, or on which tumors the research on these delivery systems has been mainly conducted. It is suggested that the authors summarize the relevant information in the discussion section.

Response. The relevant information is added to the Discussion section.

  1. I would suggest that the authors clearly list the current stage of research for these delivery systems, for example whether they are in vitro studies, or in vivo studies, or clinical trials.

Response. We have included the information on the current stage of research for the reviewed delivery systems to the additional column in all the Tables.

  1. To help the reader understand, the first occurrence of the abbreviation should be marked with the full name. For example, "ES/TRAIL" in line 300.

Response. We have added the full names at the first occurrence of the abbreviations whenever possible, and additionally provided the list of abbreviations to the end of the manuscript.

  1. Please confirm whether the content of line 341-343 and 381-384 require references.

Response. The appropriate references have been added.

  1. Please check and correct the spell that needs to be superscripted throughout the text. Such as Line 172, “Fe2+” should be “Fe2+”.

Response. The spelling and typos have been corrected throughout the manuscript.

Reviewer 2 Report

1- please make a list of abbreviations for the article.

2- please add a figure for 3D model of the receptor showing the active site.

3- Whenever possible add figures in each section  showing the interaction between ligands and the active site.

4-please move the tables to be inserted within the relevant section

Author Response

Thank you for your helpful comments.

1 - please make a list of abbreviations for the article.

Response. We have added the list of abbreviations to the end of the manuscript.

2 - please add a figure for 3D model of the receptor showing the active site.

Response. We have added a figure for 3D structure of TRAIL in complex with the DR5 receptor (Figure 1).

3 - Whenever possible add figures in each section showing the interaction between ligands and the active site.

Response. Since the majority of the reviewed nanosystems deal with the TRAIL ligand interacting with the DR5 receptor, we suppose that the Figure 1 clearly illustrates the required interaction.

4 - please move the tables to be inserted within the relevant section

Response. We have inserted the tables to the appropriate sections.
